# Neural and molecular changes during a mind-body reconceptualization, meditation, and open label placebo healing intervention
Alex Jinich-Diamant[1,2], Sierra Simpson[1], Juan P. Zuniga-Hertz[1], Ramamurthy Chitteti[1], Jan M. Schilling[1], Jacqueline A. Bonds[1], Laura Case [1], Andrei V. Chernov[1,3], Joe Dispenza[4], Jacqueline Maree[5], Natalia Esther Amkie Stahl[1], Michael Licamele[1], Narin Fazlalipour[1], Swetha Devulapalli[1], Leonardo Christov-Moore[6], Nicco Reggente [6], Michelle A. Poirier[4], Tobias Moeller-Bertram[5] & Hemal H. Patel [1,3] ✉

Mind-body interventions offer promising avenues for improving physical and mental health, yet the comprehensive biological effects of increasingly popular mind-body retreat interventions remain poorly understood. The neural and molecular effects of a 7-day retreat intervention combining meditation, reconceptualization, and open-label placebo healing rituals are investigated in an observational study on 20 healthy human participants randomly selected from 561 retreat participants. BOLD fMRI functional connectivity during rest and meditation and whole plasma proteomics, metabolomics, exosome-specific miRNA transcriptomics, and neurite growth and real-time metabolism cellular assays are compared pre- and post-intervention. Meditation decreases functional integration in the default mode ($p = 0.00009$) and salience networks ($p = 0.000003$) and decreases whole-brain modularity ($p = 0.001$). Compared to pre-intervention plasma, post plasma increases in vitro neurite outgrowth ($p = 0.01$), enhances glycolytic metabolism ($p = 0.008$), induces upregulation of BDNF ($p = 0.001$), inflammatory ($p = 0.0001$), anti-inflammatory ($p = 0.03$), and endogenous opioid ($p = 0.03$) pathways, and modulates tryptophan metabolism ($p_{FDR} = 0.03$) and neurotransmission-associated exosome miRNA transcripts. This intensive non-pharmacological mind-body intervention produces broad short-term neural and plasma-based molecular changes associated with enhanced neuroplasticity, metabolic reprogramming, and modulation of functional cell signaling pathways, highlighting the potential of mind-body techniques to modulate neural circuits and pathways important to health and well-being.

Mind-body interventions–structured practices that harness the interaction between psychological processes and physiological systems–can significantly improve human physical and mental health, yet the neural and molecular effects of increasingly popular mind-body retreat interventions remain poorly understood. In a recent RCT, reconceptualizing pain as the product of plastic brain activity rather than of peripheral tissue injury reduced pain three times more than placebo and six times more than usual treatment[1]. Placebo effects, another mind-body intervention based on healing-centered rituals, symbols, and behaviors, affect every major organ system[2,3] and sometimes surpass routine surgical outcomes[4–6]. Interestingly, open-label placebos (administered without concealment such that the subject is aware of the placebo) conserve their effectiveness against many

[1]Department of Anesthesiology, University of California San Diego, La Jolla, CA, USA. [2]Department of Cognitive Science, University of California San Diego, La Jolla, CA, USA. [3]Veterans Affairs San Diego Healthcare System, La Jolla, CA, USA. [4]Metamorphosis LLC, Ranier, WA, USA. [5]VitaMed Research, Palm Desert, CA, USA. [6]Institute for Advanced Consciousness Studies, Santa Monica, CA, USA. ✉e-mail: hepatel@health.ucsd.edu

conditions[7–9], demonstrating that placebo responses do not require deception, expectations, or conditioning. Meditation, yet another intervention based on self-regulated attentional practices, can produce subjective mystical-type experiences[10], mental health improvements[11,12], and can alter neural[13–15], immune[16], autonomic[17,18] gene expression[19–21], proteomic[22–24], and metabolomic[25,26] activity.

Each of these interventions operates through partially distinct cognitive mechanisms, raising the possibility that they may complement one another to impact the brain and body synergistically. Reconceptualization operates through conscious, discursive, and volitional alteration of core beliefs, while meditation involves a willful yet non-discursive alteration of consciousness, and open-label placebo involves conscious awareness but operates unconsciously. While research on each technique exists, their combined neural and molecular effect has never been studied. To do so, we conducted an exploratory observational study with functional magnetic resonance imaging (fMRI) and blood plasma-based high-throughput proteomics, metabolomics, exosome-specific miRNA transcriptomics; and neurite growth and real-time metabolism cellular assays (Fig. 1A) on 20 healthy adult participants (14 females, age = 46.35 ± 10.06 (SD) years) (Fig. 1B) sampled before and after a 7-day mind-body retreat (Fig. 1C). While logistical limitations prevented us from including age and gender-matched controls to isolate and mechanistically describe the neural and physiological pathways engaged by each separate mind-body technique–an equally important but separate task–, our study provides evidence for the breadth and depth of physiological effects following a holistic, multifaceted experience commonly described by participants as personally transformational.

The 7-day retreat combined lectures, meditation, and healing rituals. Daily lectures (25 total hours) emphasized the body's self-healing abilities, the mind's capacity to shape lived reality, and the healing power of present-centeredness and mystical-type experiences. All meditations (33 total hours) were guided, delivered with atmospheric music, and taught Kundalini techniques, which combine conscious meta-awareness and conscious breathing exercises with slow, ascending, focused interoceptive attention on purported energetic centers along the midline (e.g., brow, throat, heart) which, according to practitioners, can reprocess embodied trauma and catalyze adaptive mental and physical changes[27,28]. Guidance also emphasized sustaining a heart-centered state devoid of thinking or judgment and focusing awareness on a void beyond one's normal sense of space and time —a common theme in some contemplative practices[29]. Guided healing rituals (5 total hours) brought 6–8 "healers" around one "healee" in which the former were instructed to practice loving-kindness compassion meditation while focusing attention on their heart, hands, and on the latter's body. A healing mechanism was not presented, but the possibility that healing could occur on either party because of the ritual was mentioned, similarly to how open-label placebos are presented in trials[30]. All study subjects participated as healers. Of the 20 participants, 11 were "advanced" meditators who had practiced the techniques taught for at least six months, while 9 were "novices" who had not. No pharmacological substances, including any psychedelics, were involved in the retreat.

We characterized the resting and meditation states experienced by participants pre- and post-intervention. Consistent with previous meditation studies[31,32], we observed higher meditation-associated whole-brain

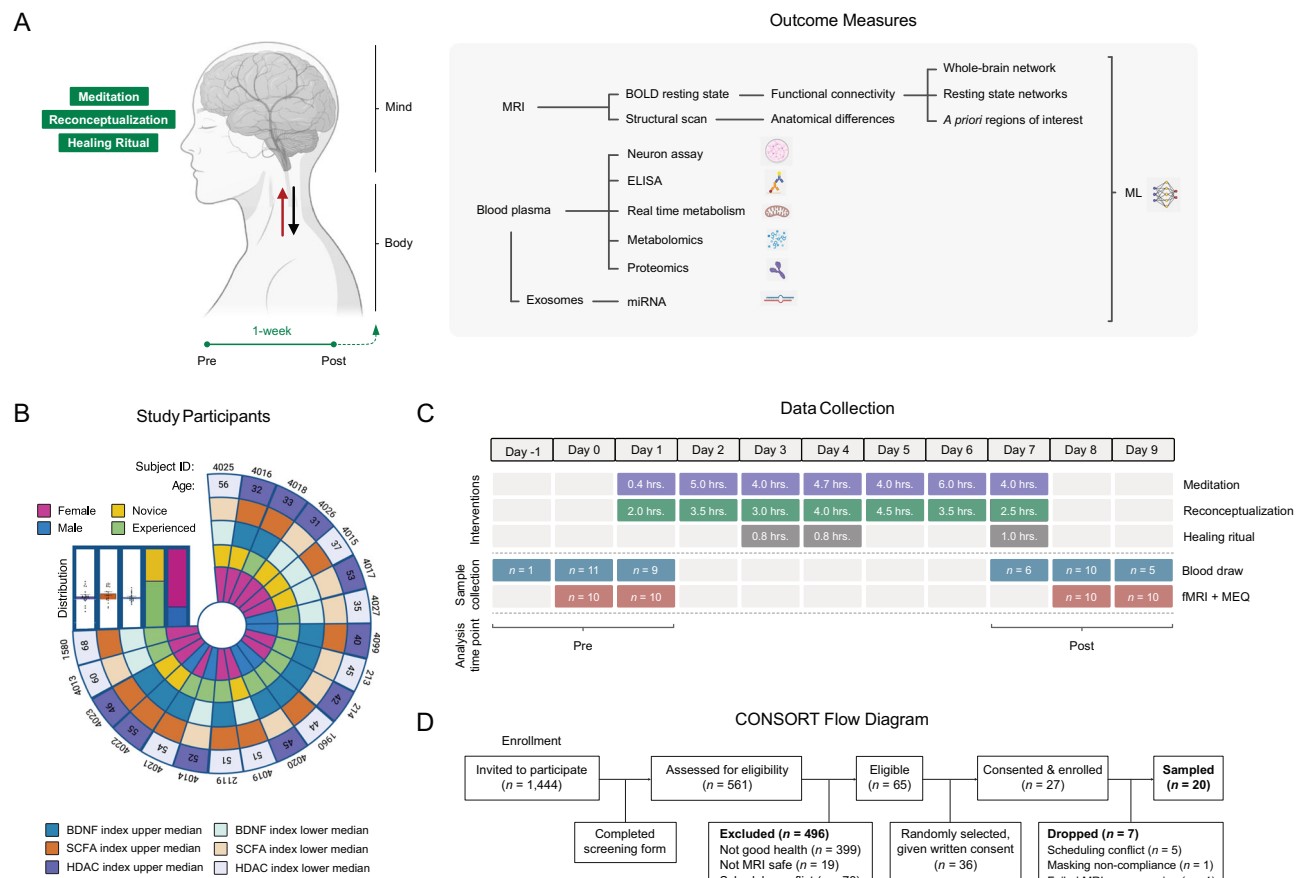

**Fig. 1 | Study design, participants, data collection, and recruitment. A** Outcome measures to capture biological changes associated with brain and body. Created with BioRender. Simpson, S., Jinich, A. (2025). BioRender.com/ryzs1cd. **B** Participant characteristics, including age, subject, gender, meditation experience level, and top/ bottom median scores for BDNF, short-chain fatty acid (SCFA) metabolism, and HDAC proteomic pathways. **C** Intervention and data collection timeline. (fMRI functional magnetic resonance imaging, MEQ mystical experience questionnaire). **D** CONSORT flow diagram.

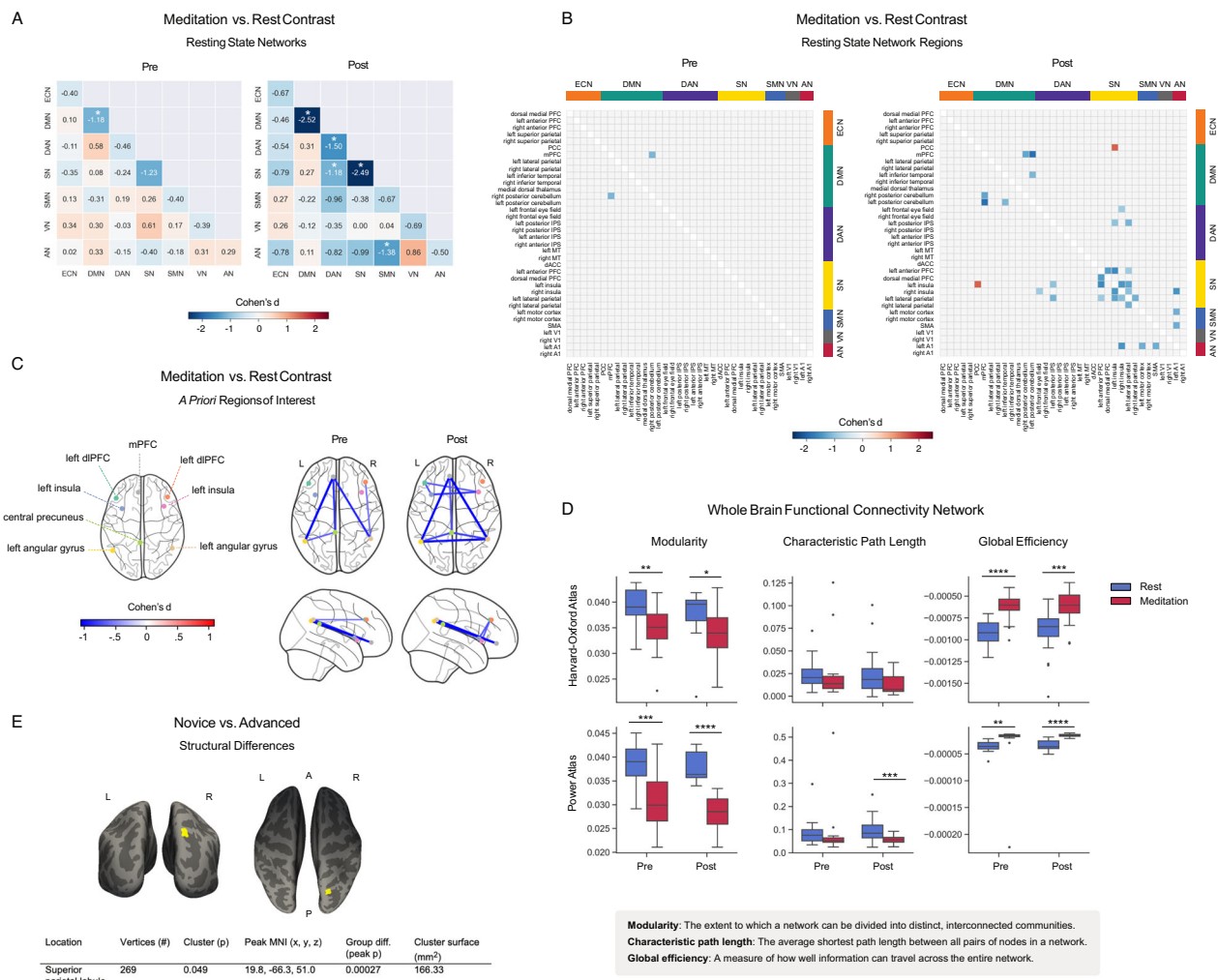

**Fig. 2 | Functional magnetic resonance imaging (fMRI) (n = 19 participants).**
**A** Meditation vs. rest RSN functional connectivity pre- and post-intervention. Cell values and color scale represent Cohen's *d* effect size values from paired *t*-tests. Asterisks denote $p_{FWE} < 0.05$ significance (28 multiple comparisons). ECN (executive control network), DMN (default mode network), DAN (dorsal attention network), SN (salience network), SMN (somatomotor network), VN (visual network), and AN (auditory network). **B** Meditation vs. rest RSN-region functional connectivity pre- and post-intervention. Cell values and color scale represent Cohen's *d* effect size values from paired *t*-tests. Asterisks denote $p_{FWE} < 0.05$ (630 multiple comparisons). **C** Significant ($p < 0.05$) meditation-induced functional connectivity changes between a priori ROIs pre- and post-intervention. Color scale represents Cohen's *d* effect sizes from paired *t*-tests. **D** Whole-cortex/brain network measures for 48-region Harvard-Oxford cortical atlas and Power (2011) 264-region brain atlas (n = 19). Asterisks denote * ($p < 0.05$), ** ($p < 0.01$), *** ($p < 0.001$), **** ($p < 0.0001$). **E** Anatomical Structure. Coronal and axial views with superior parietal lobule cluster (yellow), where advanced practitioners had greater gray matter volume than novices at baseline.

functional integration and reduced intra-network connectivity in the default mode and salience networks. Comparing pre- to post-intervention whole plasma, we report evidence of enhanced neuroplasticity and glycolytic metabolism, as well as activation of endogenous opioid and neuromodulatory pathways.

## Results
### Functional brain activity
To characterize the neural signature of the meditative state, participants underwent structural and blood-oxygenation-level-dependent (BOLD) functional MRI scans during rest (5 min) and meditation (15 min). Mystical experience questionnaire (MEQ) scores reflective of the scanner meditation increased significantly following the intervention (n = 20; pre = 2.37 ± 1.26 (*SD*); post = 3.02 ± 1.44) (Wilcoxon signed rank test W = 35.0, $p = 0.03$), indicating a deepening of the meditative state between sessions. One participant's data was excluded from further analysis due to corrupt T1w data. Participants moved more during meditation than rest, a potential confound revealed by the significant effect of task (meditation, rest) on mean

framewise displacement on a two-way (task × time) repeated measures ANOVA (n = 19, $F_{(1,18)} = 25.1$, $p = 0.00009$, $\eta^2 p = 0.58$) (Supplementary Tables 1 and 2).

### Meditation versus rest
We examined functional connectivity in seven resting state networks (RSNs) (Fig. 2A, B), eight regions of interest (ROIs) (Fig. 2C) (coordinates in Supplementary Tables 3 and 4), and two whole-brain networks (Fig. 2D), comparing meditation with rest (n = 19, $p_{FWE} = 0.0018$ for 28 network pairs). All resting state networks had higher intra- than inter-network connectivity (Supplementary Fig. 1A). Compared to rest, meditation reduced intra-network connectivity in the salience network pre- and post-intervention (pre: $t = -4.32$, $p = 0.0004$, Cohen's $d = -0.87$; post: $t = -6.71$, $p = 0.000003$, Cohen's $d = -1.76$), default mode network post-intervention (DMN) ($t = -5.04$, $p = 0.00009$, Cohen's $d = -1.78$), and dorsal attention network (DAN) post-intervention ($t = -3.71$, $p = 0.002$, Cohen's $d = -1.06$) (Fig. 2A). Meditation also showed reduced inter-network connectivity post-intervention between the salience and dorsal attention

**Table 1 | Statistically significant ($p_{FWE}$ < 0.05) rest-vs-meditation differences on resting state network regions ($n$ = 19 participants)**

| Session | RSN component region pair | | RSN(s) | $t$ | $p$ | Cohen's $d$ |
|---|---|---|---|---|---|---|
| Pre | R post. Cerebellum | Medial PFC | DMN | −4.44 | 0.0003 | −1.15 |
| Post | R post. Cerebellum | Medial PFC | DMN | −5.81 | 0.00002 | −1.30 |
| Post | L post. Cerebellum | Medial PFC | DMN | −6.30 | 0.000006 | −1.87 |
| Post | L post. Cerebellum | L inferior temporal | DMN | −4.56 | 0.0002 | −1.13 |
| Post | R anterior PFC | L anterior PFC | SN | −7.41 | 0.0000007 | −1.38 |
| Post | L insula | L posterior IPS | SN | −4.32 | 0.0004 | −1.01 |
| Post | L insula | L anterior PFC | SN | −5.30 | 0.00005 | −1.57 |
| Post | R insula | L insula | SN | −5.58 | 0.00003 | −1.44 |
| Post | L lateral parietal | L anterior PFC | SN | −4.62 | 0.0004 | −0.88 |
| Post | L lateral parietal | L insula | SN | −7.30 | 0.0000009 | −1.30 |
| Post | L lateral parietal | R insula | SN | −6.01 | 0.00001 | −1.02 |
| Post | R lateral parietal | L lateral parietal | SN | −4.41 | 0.0003 | −1.18 |
| Post | L insula | PCC | SN-DMN | 4.50 | 0.0003 | 1.47 |
| Post | R insula | L frontal eye field | SN-DAN | −4.24 | 0.0005 | −0.89 |
| Post | L lateral parietal | L posterior IPS | SN-DAN | −4.66 | 0.0002 | −1.15 |
| Post | L A1 | R insula | AN-SN | −4.68 | 0.0002 | −1.47 |
| Post | L A1 | L motor cortex | AN-SMN | −4.67 | 0.0002 | −1.19 |
| Post | L A1 | SMA | AN-SMN | −4.55 | 0.0002 | −1.22 |

**Table 2 | Significant ($p_{FDR}$ < 0.05) pre-to-post differences on RSN component regions ($n$ = 19 participants)**

| Task | RSN Component Region Pair | | RSN(s) | $t$ | $p$ | Cohen's $d$ |
|---|---|---|---|---|---|---|
| meditation | L posterior IPS | L A1 | DAN-AN | −4.72 | 0.0002 | −1.01 |
| meditation | L posterior IPS | L lateral parietal | DAN-SN | −4.44 | 0.0003 | −0.80 |

networks ($t = −4.09$, $p = 0.0007$, Cohen's $d = −1.18$) and between the somatomotor and auditory networks ($t = −5.30$, $p = 0.00005$, Cohen's $d = −1.38$).

For individual network-component regions (Fig. 2B and Table 1), both pre- and post-intervention, meditation reduced connectivity between the right posterior cerebellum and medial prefrontal cortex (pre: $t = −4.44$, $p = 0.0003$, Cohen's $d = −1.15$; post: $t = −5.81$, $p = 0.00002$, Cohen's $d = −1.30$) and bilateral connectivity between insular cortices, lateral parietal cortices, and anterior prefrontal cortices (all $p \leq 0.0003$; Table 1), network hubs associated with sensory, affective, and cognitive functions that shape the prediction-driven conscious state[33]. Interestingly, the only meditation-driven connectivity increase was post-intervention between the left insula and posterior cingulate cortex ($t = 4.50$, $p = 0.0003$, Cohen's $d = 1.47$), a finding previously reported during absorptive trance states[34]. ROIs also revealed lower functional connectivity between key network hubs, including medial prefrontal cortex (mPFC), precuneus, bilateral insular cortices, and bilateral angular gyri (Supplementary Table 5), serving as an internal replication and returning results agreeable with the RSN data-driven approach.

To observe meditation-induced changes at the whole-brain/cortex level, we parcellated the brain into 48 cortical regions using the Harvard-Oxford atlas[35–37] and into 264 brain regions using the Power atlas[38] and calculated whole-brain/cortex network measures (Supplementary Table 6). For the cortical parcellation, a 2 (pre/post) by 2 (rest/meditation) repeated measures ANOVA (Supplementary Table 7) ($n = 19$) revealed a significant effect of meditation on network modularity ($F_{(1,18)} = 15$, $p = 0.001$, $\eta^2 p = 0.45$) and global efficiency ($F_{(1,18)} = 48$, $p = 0.000002$, $\eta^2 p = 0.73$), and no significant effect on characteristic path length, with similarly significant whole-brain results. Post-hoc Wilcoxon signed rank tests confirmed that meditation decreased modularity (pre: $W = 18.0$, $p = 0.001$, Cohen's $d = −1.07$; post: $W = 45.0$, $p = 0.04$, Cohen's $d = −0.77$) and increased global efficiency (pre: $W = 3.0$, $p = 0.00002$, Cohen's $d = 2.04$; post: $W = 10.0$, $p = 0.0002$, Cohen's $d = 1.26$) as compared to rest (Fig. 2D). These results were robust to excluding BOLD runs with mean framewise displacement > 0.3 mm, indicating they were not due to higher meditation-associated head motion.

Meditation thus reduced functional connectivity in the default mode and salience networks and induced a whole-brain functional state less segregated into distinct modules, that allows for more efficient information flow.

### Pre versus post

Pre- to post-intervention, functional connectivity during meditation decreased between executive control and salience networks ($t = −2.61$, $p = 0.02$, Cohen's $d = −0.62$) but did not survive correction for multiple comparisons ($p_{FWE} = 0.0018$) (Supplementary Fig. 1B). For network component regions, connectivity during meditation decreased significantly pre-to-post between the left posterior intra-parietal sulcus and auditory cortex and between the left posterior intra-parietal sulcus and left lateral parietal cortex (Table 2). No significant changes were found between a priori ROIs (Supplementary Table 8).

### Neuroanatomical differences

No significant anatomical changes were observed pre-to-post intervention, but advanced practitioners showed greater gray matter volume in the right superior parietal lobule at baseline ($n = 19$, $p = 0.049$) (Fig. 2E), a region linked to spatial awareness and body representation, which has been reported to have greater cortical thickness in experienced Zen meditators[39]. This difference warrants further investigation with larger samples and comparisons with age-matched non-meditator controls.

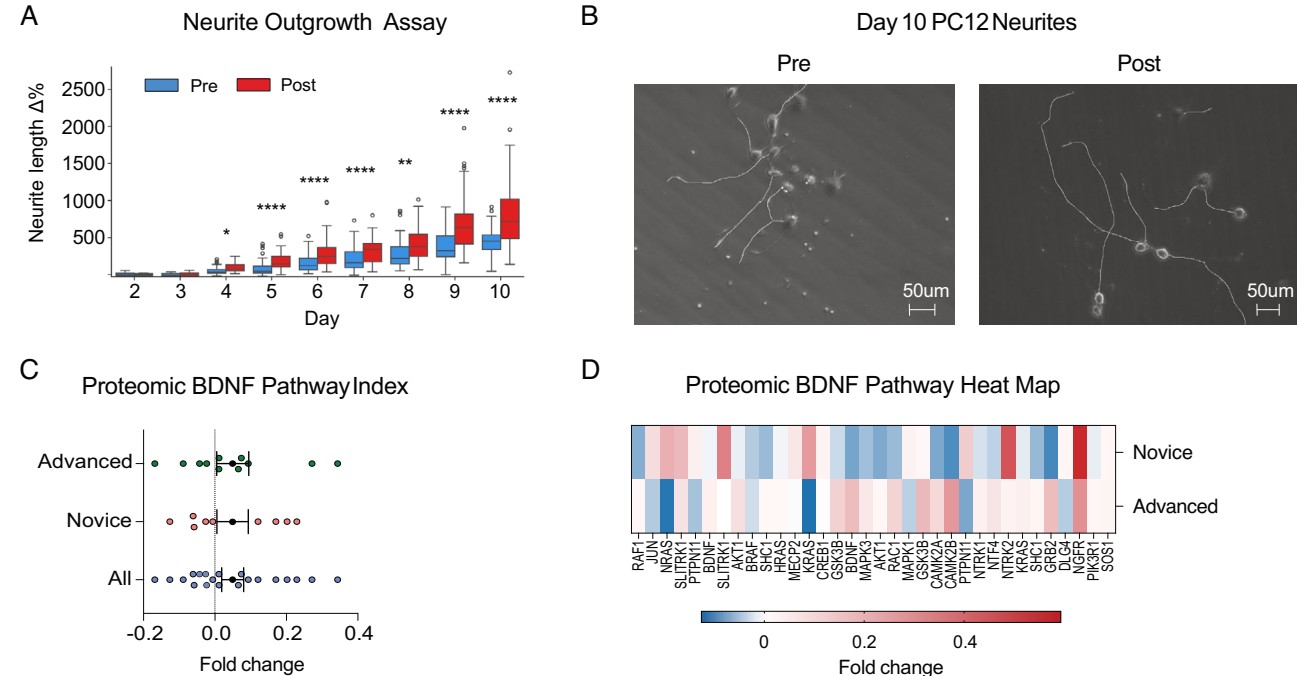

**Fig. 3 | Neuronal growth ($n = 20$ participants). A** PC12 neurite growth: Percent change in mean neurite length from baseline (day 2 post-NGF treatment). Asterisks denote * ($p < 0.05$), ** ($p < 0.01$), *** ($p < 0.001$), **** ($p < 0.0001$). **B** Phase contrast 20× microscopy images of PC12 cells on day 10 post-NGF treated with pre- and post-intervention plasma with longest neurite per cell body traced. **C** BDNF pathway index pre-to-post fold change. Error bars denote SEM. Fold change levels significantly above zero signal upregulated pathway or protein cluster. **D** BDNF Pathway Heat Map. Fold change per protein for advanced and novice participants.

## Whole plasma

Having characterized the neural state induced by meditation, we turned our attention to the broader metabolic and molecular changes in whole plasma.

## Enhanced neuroplasticity

Participants' anecdotal reports consistently emphasize radical psychological breakthroughs, and previous meditation studies have reported increased BDNF (brain-derived neurotrophic factor) levels[22] consistent with enhanced neuroplasticity. To investigate whether the intervention affected circulating plasma factors conducive to neuroplasticity, we treated cultured glutamatergic PC12 neuroendocrine cells with NGF (nerve growth factor) and 1% pre- and post-intervention plasma and quantified neurite outgrowth length. Starting on day 4 post-NGF treatment, post-plasma-treated cells exhibited significantly longer neurites than pre-plasma-treated cells ($t = -2.52$, $p = 0.01$, Cohen's $d = 0.59$) and continued to do so until the end of the experiment (Fig. 3A, B).

To investigate proteomic factors driving this effect, we quantitatively measured 7596 protein targets using the high-throughput SomaScan assay and constructed a BDNF pathway 26-protein pre-to-post foldchange index (Fig. 3C, D), which was significantly upregulated (1-sample $t$-test: $t = 3.21$, $p = 0.001$, Cohen's $d = 0.12$). While BDNF itself was not significantly affected, SLITRK1 (SLIT and NTRK-like family member 1), a protein that promotes excitatory synapse development and glutamatergic neurite outgrowth[40,41], increased significantly pre-to-post ($W = 39.0$, $p = 0.01$, CLES = 0.35). NGFR (nerve growth factor receptor), a TNF receptor that binds to NGF and BDNF and plays an essential role in neural cell differentiation and survival, also increased ($W = 58.0$, $p = 0.08$, CLES = 0.35).

## Metabolic reprogramming

Previous studies have characterized meditation as a hypometabolic state[42] and reported enhanced glycolysis in Tibetan Buddhist monks[24]. To test the intervention's effect on real-time metabolism, we treated BE(2)M17 human neuroblastoma cells with 1% plasma from all participants for 60 min and performed Seahorse XF assays. An ATP Rate assay (Fig. 4A) revealed that changes in plasma produced during the intervention induced a compensatory shift from mitochondrial to glycolytic ATP production, with post-plasma-treated cells showing significantly higher basal glycolytic rate than pre-plasma-treated cells ($t = 2.95$, $p = 0.008$, Cohen's $d = 0.36$) despite no significant differences in mitochondrial ($t = -1.04$, $p = 0.31$, Cohen's $d = -0.28$) or total ATP production rate ($t = 1.46$, $p = 0.16$, Cohen's $d = 0.25$). A glycolytic rate assay (Fig. 4B) confirmed that, compared to pre-plasma, post-plasma increased basal glycolytic rate ($t = 3.39$, $p = 0.003$, Cohen's $d = 0.63$) and lowered basal respiration rate ($t = -4.80$, $p = 0.0001$, Cohen's $d = -1.26$). Mitochondrial stress parameters (Fig. 4C) did not significantly differ between cells exposed to pre- and post-plasma, while, compared to cells exposed to growth medium only, plasma-exposed cells displayed significantly higher ATP production and glycolytic rates.

As before, to see if the plasma proteome reflected these changes, we pre-selected 19 proteins involved in glycolysis and oxidative phosphorylation and calculated pre-to-post foldchanges and an index average (Fig. 4D–G). The glycolysis index increased significantly pre-to-post ($t = 3.37$, $p = 0.0008$, Cohen's $d = 0.23$), with 12 upregulated targets led by ENO2 (enolase 2, a neuron-specific converter of 2-phosphoglycerate into phosphoenolpyruvate) ($W = 48.0$, $p = 0.03$, CLES = 0.33) and LDHA (lactate dehydrogenase A, converter of pyruvate into lactate in anaerobic glycolysis ($W = 38.0$, $p = 0.01$, CLES = 0.22). Oxidative phosphorylation associated proteins trended upwards ($t = 1.38$, $p = 0.17$, Cohen's $d = 0.13$) but did not reach statistical significance.

## Functional cellular signaling

Having investigated specific pathways of interest, we performed an exploratory, hypothesis-free analysis of proteomic and metabolomic results.

## Proteomics

Volcano plot analysis (Fig. 5A) revealed 21 significantly altered proteins. Cofilin-2 (COF2) and Enoyl-CoA hydratase were significantly upregulated, which suggests enhanced cellular processes related to cytoskeletal regulation and fatty acid metabolism. IL1-F6 (interleukin-36 alpha), MYPC1 (myosin

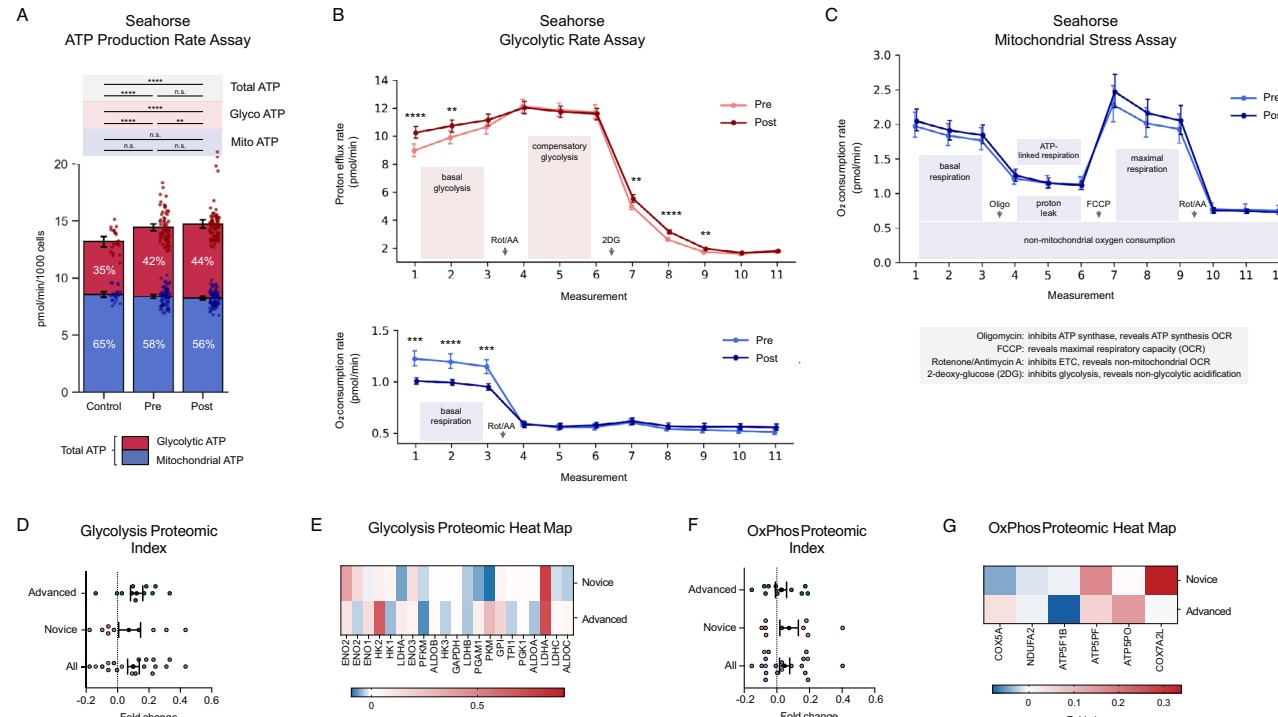

**Fig. 4 | Metabolic effects ($n = 20$ participants). A–C** Seahorse XF analyzer real time cellular metabolic assays with BE(2)M17 human neuroblastoma cells treated with 1% pre- and post-retreat plasma for 60-min or assay buffer (control). Mean ± 95% CI error bars. **A** ATP production rate assay with total ATP production rate (mean ± 95% CI) broken into glycolytic and mitochondrial ATP production rates. **B** Glycolytic rate assay, including basal and compensatory glycolytic rates (top) and basal mitochondrial respiration rate (bottom). Mean ± 95% CI error bars. **C** Mitochondrial respiration stress assay. Mean ± 95% CI error bars. Asterisks

denote * ($p < 0.05$), ** ($p < 0.01$), *** ($p < 0.001$), **** ($p < 0.0001$). **D** Glycolysis pathway index pre-to-post fold change. Error bars denote SEM. Index fold change levels significantly above zero signal an upregulated pathway or protein cluster. **E** Glycolysis pathway heat map. Fold change level per component protein for advanced and novice participants. **F, G** Oxidative phosphorylation pathway index with pre-to-post fold change (**F**) and heat map for advanced and novice participants (**G**). Error bars denote SEM.

binding protein C1), LDHA, and FGF-19 (fibroblast growth factor 19) exhibited moderate upregulation.

Protein-protein interaction networks (Fig. 5B) revealed three significantly altered protein clusters related to mitochondrial energy production (ATP5F1, ATP6V1F); fatty acid metabolism (ECHS1, POP7, and ACAT2); nucleosome organization (HDAC1, RANGAP1, RBL2, C3, and MYOM2).

Pathway enrichment analysis (Fig. 5C) showed upregulated proteins associated with muscle cell apoptosis, amyloid precursor protein catabolism, and mitochondrial proton transport pathways ($p_{adjusted} < 0.05$), suggesting alterations in metabolic pathways and muscle-related cellular processes. Interestingly, pathways related to butanoate, propanoate metabolism, and tryptophan biosynthesis were enriched, suggesting shifts in key metabolic routes. (See Supplementary Figs. 3 and 4 for g:Profiler enrichment and pathway heatmaps.)

We also created proteomic indices to examine inflammation, epigenetic regulation, and short-chain fatty acid (SCFA) metabolism pathways, comparing across timepoint (pre/post) and experience level (novice/advanced) (Fig. 5E) (Supplementary Fig. 5). The epigenetic (HDAC) index demonstrated elevated sirtuins and HDAC expression in the advanced group, reflecting increased epigenetic regulation and possible stress resilience. In contrast, the novice group showed differences in mitochondrial function and chromatin remodeling. SCFA index showed increased expression of acyl-CoA dehydrogenase short-chain (ACADS) and fatty acid synthase (FASN) in the advanced group, suggesting a shift toward enhanced fatty acid oxidation and improved metabolic efficiency.

## Inflammation, anti-inflammation, and cellular turnover

To assess whether the intervention elicited inflammatory or anti-inflammatory cascades, we examined a panel of 23 inflammatory and 21 anti-inflammatory proteins (Fig. 5E). We found significant upregulation of inflammatory markers ($t = 3.81$, $p = 0.0001$, Cohen's $d = 0.15$), driven by increases in S100A8 (calgranulin A) ($W = 30.0$, $p = 0.004$, CLES = 0.33) and CCL2 (C-C motif chemokine 2) ($W = 47.0$, $p = 0.03$, CLES = 0.33) and trending increases in IL-6, S100A9 (S100 calcium-binding protein A9; calgranulin B), and PTGS2 (prostaglandin G/COX-2). These findings align with the role of S100A8 and S100A9 as alarmins—endogenous molecules released in response to cellular damage or stress known to induce secretion of inflammatory mediators IL-6, IL-8, and CCL2[43,44].

Interestingly, we also observed a significantly upregulated anti-inflammatory markers index ($t = 2.25$, $p = 0.03$, Cohen's $d = 0.09$), with positively trending levels of TGF-b1 (transforming growth factor beta-1), NFKBIA (NF-kappa-B inhibitor), STAT6 (signal transducer and activator of transcription 6), CEBPB (CCAAT/enhancer-binding protein beta), IL1, SOCS3 (suppressor of cytokine signaling 3), and TNFAIP3 (TNF alpha-induced protein 3). Concurrent activation of both pathways suggests a dynamic process of immune modulation, possibly reflecting enhanced cellular turnover or repair mechanisms. We also measured plasma nanoparticles and found no significant change in total nanoparticle concentration ($t = -0.17$, $p = 0.87$, Cohen's $d = -0.03$), but did find a significant decrease in the percentage of particles in the exosome range (20–120 nm diameter) ($t = -2.09$, $p = 0.04$, Cohen's $d = -0.65$) (Supplementary Fig. 2A, B) consistent with both higher cellular turnover and metabolic suppression.

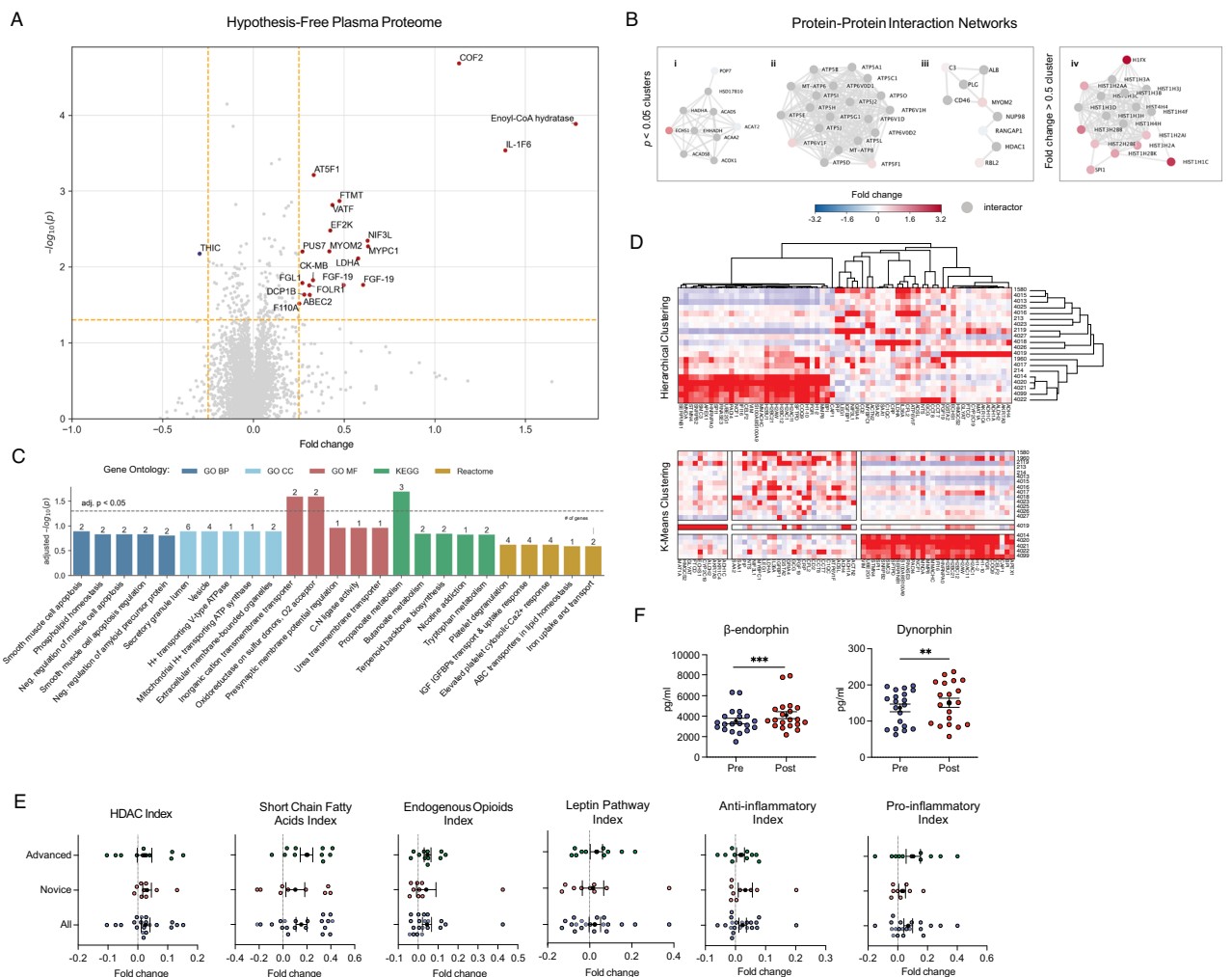

**Fig. 5 | Plasma proteome (n = 20 participants). A Plasma Proteome.** Volcano plot of pre/post fold changes (x-axis) and paired t-test –log(p-values) (y-axis) for 7596 proteins. Vertical yellow dashed lines mark a linear fold change of 0.25 in either direction; horizontal yellow dashed line marks p = 0.05 cutoff. Proteins in top right/left quadrants are significantly up/downregulated. **B Protein-Protein Interactions.** Heat-diffused network clusters with ≤ 20 interactors for proteins with pre/post (i–iii) p < 0.05 and (iv) |foldchange| > 0.5. Top gene enrichment terms for each cluster are: (i) fatty acid metabolism, (ii) proton-transporting two-sector ATPase complex, (iii) complement system, and (iv) nucleosome organization. **C Gene Enrichment.** Enrichment analysis (Enrichr) with pre/post significantly altered proteins (p < 0.01) and assay targets as background. **D Proteome Cluster Analysis.** Hierarchical and K-means clustering of |fold changes| > 0.5 proteins. Dendrogram branch lengths indicate degree of similarity between expression profiles. Nodes represent protein targets; colors indicate fold change. **E Proteomic A Priori Gene Indices.** Mean pre/post fold change for protein clusters or pathways. Index proteins in Supplementary Table 15. Mean ± SEM error bars. **F ELISA Assay Biomarkers.** Mean ± 95% CI error bars. Asterisks denote * (p < 0.05), ** (p < 0.01), *** (p < 0.001).

## Endogenous opioids

Placebo effects are known to engage endogenous opioid and endocrine systems. We assessed levels of 15 proteomic targets within the endogenous opioid pathway and found the pathway index to be significantly upregulated (t = 2.14, p = 0.03, Cohen's d = 0.12), driven by positively trending increases in opioid peptide precursor PENK (proenkephalin-A), opioid peptide PDYN (dynorphin A), GABR2:CD (GABBA B receptor subunit 2: cytoplasmic domain), and CAMK2B (calcium/calmodulin-dependent protein kinase type II), which is known to increase after opioid administration (Fig. 5E). ELISA assays to confirm these findings (Fig. 5F and Supplementary Table 11) revealed significant increases of beta-endorphin (W = 14.0, p = 0.0002, Cohen's d = 0.42) and dynorphin (W = 37.0, p = 0.009, Cohen's d = 0.27).

## Metabolomics

We performed liquid chromatography-mass spectrometry-based metabolomic analysis on plasma. Partial least squares discriminant analysis

(PLS-DA) (Fig. 6A) revealed distinct metabolic profiles between time points, with key metabolites involved in synaptic plasticity, metabolism, RNA modulation, neurotransmitter availability, and inflammation and over 25 metabolites with VIP (variable importance in projection) scores > 2 (Fig. 6B, C).

MetaboAnalyst pathway enrichment detected 53 impacted pathways (Fig. 6D and Supplementary Table 16). Six showed significant pre-to-post changes (p < 0.05), led by tryptophan metabolism ($p_{FDR}$ = 0.03) (Fig. 6E and Supplementary Fig. 6), with decreases in upstream and downstream metabolites, including L-tryptophan (p = 0.03), tryptamine (p = 0.04), L-kynurenine (p = 0.04), indole-3-acetate (p = 0.001), and 5-methoxyindoleacetate (p = 0.001). The steroid hormone biosynthesis pathway showed lower androstenediol (p = 0.04), testosterone (p = 0.19), cortisol (p = 0.06), cortisone (p = 0.07), 21-deoxycortisol (p = 0.77), corticosterone (p = 0.07), and 11-dehydrocorticosterone (p = 0.02). Lower cortisol levels reported in previous meditation studies were interpreted as signs of improved stress response regulation[45].

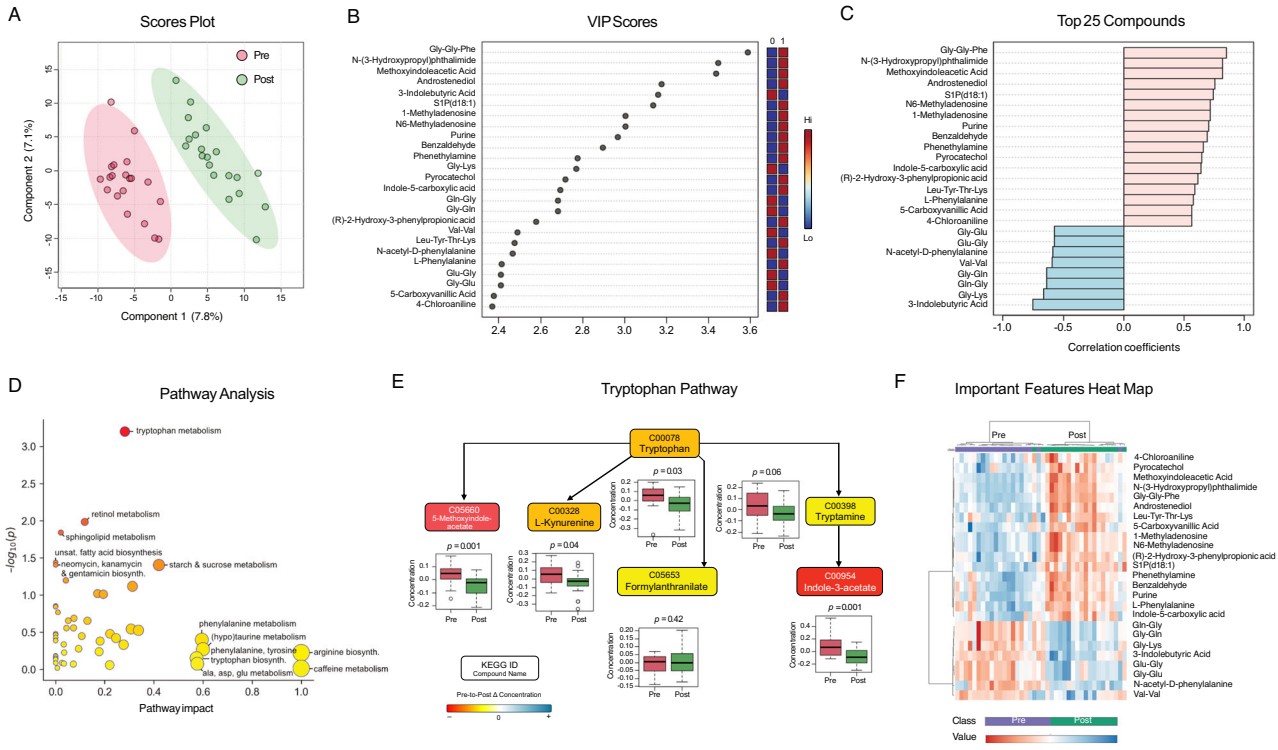

**Fig. 6 | Metabolomics (*n* = 20 participants). A PLS-DA Scores Plot**. Scatter plot shows 2 principal components with greatest variation. Ovals show 95% confidence intervals. Similar observations cluster together. Component 1 (*x*-axis) contains 7.8% of total variation; component 2 contains 7.1%. Oval cluster spatial separation indicates systematic differences between timepoints. **B PLS-DA Variable Importance in Projection (VIP) Scores**. VIP scores indicate metabolite's contribution to timepoint separation, ranked in descending order. Top metabolites are involved in neurotransmitter regulation, lipid signaling, and RNA modification, suggesting the intervention induced broad metabolic and molecular adaptations. **C Top PLS-DA Compounds**. Correlation coefficients between metabolites and PLS-DA discriminant function. Each metabolite's correlation coefficient indicates strength and direction of association with timepoint separation (positive = direct, negative = inverse, closer to ± 1 = more influential in driving group separation). Top

metabolites highlight changes in neurotransmitter precursors, lipid signaling, and amino acid metabolism. **D Pathway Analysis**. Metabolite pathways' impact scores (*x*-axis) and *p*-values (*y*-axis, -log₁₀ transformed). Larger impact values indicate greater contribution to the pathway. Bubble size reflects the pathway's enrichment score, and bubble color corresponds to significance level (darker indicating lower *p*-value). **E Tryptophan Pathway**. Colored boxes represent detected metabolites. Color scale represents magnitude of pre-to-post change. Box plots show pre and post concentration distributions and paired *t*-test values with raw *p*-values. **F Important Features Heat Map** showing top differentially expressed metabolites, including several phenylalanine-related metabolites involved in neurotransmitter synthesis and metabolic regulation. Color scale indicates relative abundance (red = higher; blue = lower). Clustering highlights metabolic pathway shifts, including phenylalanine metabolism and lipid signaling.

Among top-ranking metabolites, three were related to phenylalanine metabolism, including CAN-2-Hydroxy-3-phenylpropionic acid, N-acetyl-D-phenylalanine, and phenethylamine, which can act as neurotransmitter storage, contribute to stress adaptation, and enhance dopamine, norepinephrine, and serotonin release. Two RNA-related metabolites, 1-methyladenosine and N6-methyladenosine, were linked to inflammatory responses, with roles in RNA metabolism, methylation, and cellular signaling. 3-indolebutyric acid, associated with tryptophan metabolism, may influence neurotransmitter synthesis and synaptic plasticity. Several dipeptides were identified as enhancers of gut microbial activity, promoting SCFA production, which is crucial for regulating inflammation and immune function. S1P (d18:1), a signaling lipid, may reflect gut barrier integrity and inflammatory pathway alterations. Figure 6F shows a distinct clustering of metabolites by relative abundance, highlighting these key contributors.

**Exosome-specific transcriptomics**
We analyzed differentially expressed exosome-specific extracellular microRNAs, non-coding RNAs, and RNAs mapping to protein-coding mRNAs. Data from 6 participants was excluded at preprocessing (*n* = 16). At least 18 non-coding exRNAs (*p* < 0.1, log₂FC > ±0.58) exhibited distinct expression profiles on pre and post timepoints (Fig. 7A, B). Principal components (PC) explain 46.3% of the total timepoint variation (Fig. 7C),

indicating significant shifts in non-coding exRNA expression during the intervention. At least 5.99% of the variance was attributed to experience levels, with partial separation between novice and advanced meditators (Fig. 7D). Correlations between principal components and key variables such as timepoint and experience reveal that PC1 and PC2 strongly correlate with timepoint (*R*² = 0.35 and 0.28, respectively), while PC9 moderately correlates with experience (*R*² = 0.31), indicating that both influence exRNA expression (Fig. 7E).

We also identified exRNAs mapped to at least 66 annotated protein-coding mRNAs upregulated or downregulated post-intervention (Fig. 7F, G). Some of the upregulated exRNAs could be mapped to ras/rab interactor 1 (*RIBC1*), synapsin 3 (*SYN3*), and glutamate ionotropic receptor kainite type subunit 3 (*GRIK3*), genes linked to synaptic function and neurotransmission. The enrichment of exRNAs specific to solute carrier family 27 member 1 (*SLC27A1*) and proprotein convertase subtilisin/kexin type 9 (*PCSK9*) could enhance neural signaling and metabolic regulation. Downstream functional analysis of exRNAs in pre-vs-post exosome fractions against the Reactome database[46] (Fig. 7H) predicted pathways related to neurotransmission (Serotonin/Dopamine Neurotransmitter Release Cycle), further highlighting the enhancement of synaptic activity post-intervention. Metabolic pathways, including Transport of Vitamins and Nucleosides, could reflect broader metabolic changes.

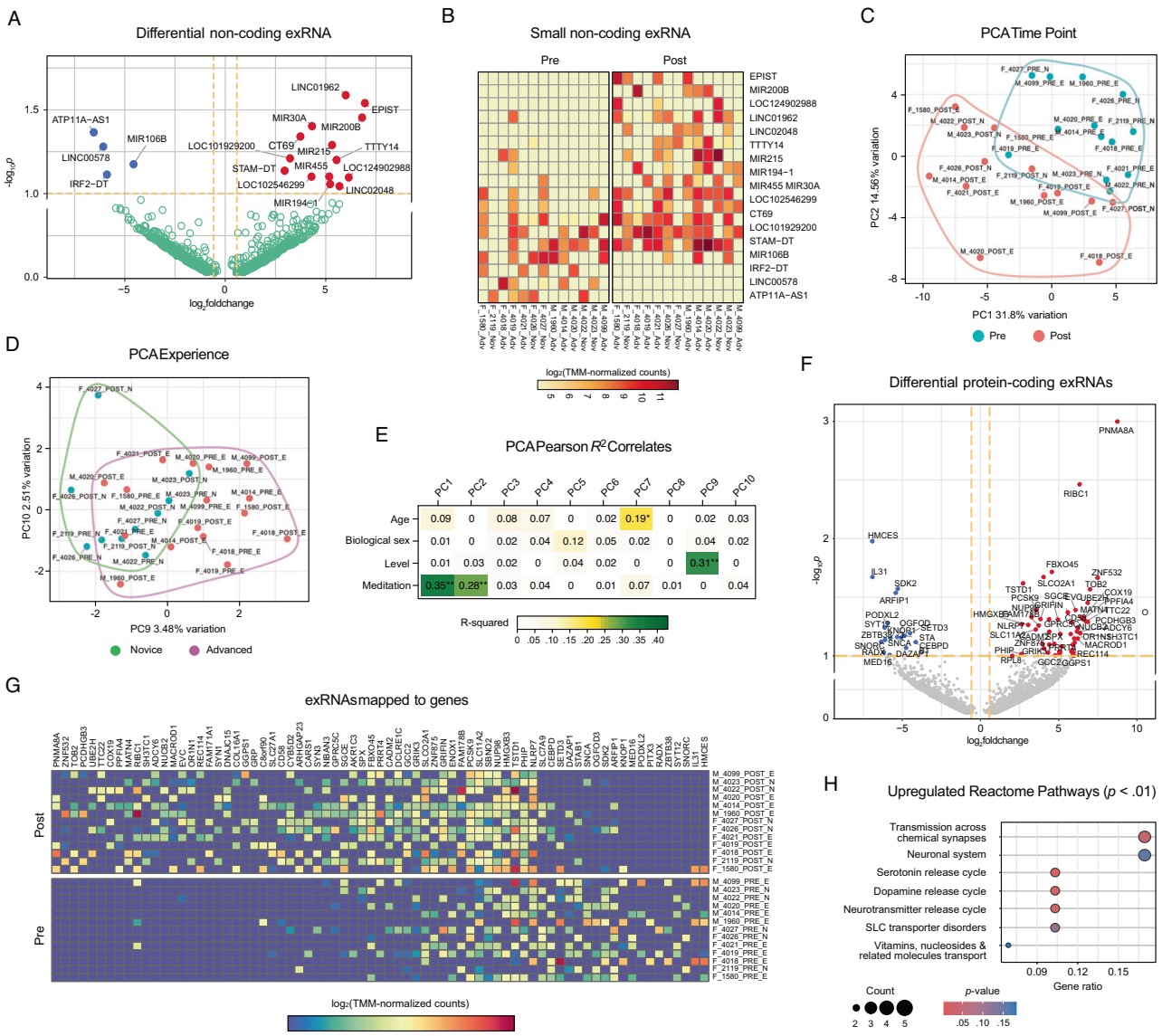

**Fig. 7 | Exosome transcriptomics (_n_ = 16 participants). A** Differential non-coding exRNAs (miRNAs, ncRNAs) on volcano plot of -log$_{10}$p versus log$_2$FC (pre/post difference). Blue/red dots denote exRNAs prevalent in pre/post-intervention exosomes, respectively; green dots denote exRNAs without significant difference. Dashed orange lines indicate $p < 0.1$ and log$_2$FC > ±0.58 thresholds. Full list of exRNAs on Supplementary Table 17. **B** Differential expression (log$_2$-transformed TMM-normalized counts) of 18 non-coding exRNAs. Colors indicate counts. Sample name: (M male, F female)_participantID_(pre; post)_(E, advanced; N, novice). **C, D** Principal component analysis of top normalized exRNA counts. Encircled are clusters corresponding to **C** pre/post, and **D** novice/advanced. **E** Eigenvector plot showing the correlation of principal components to variables'

metadata and test significances (** denotes $p < 0.01$; * denotes $p < 0.05$). Values correspond to Pearson $R^2$ values and are colored by significance. **F** Differential protein-coding exRNAs (excluding ncRNAs) on volcano scatter plot of -log$_{10}$p versus log$_2$FC (pre/post difference). Blue/red dots denote exRNAs prevalent in pre/post exosomes, respectively; green dots denote exRNAs without significant difference. Dashed orange lines indicate $p < 0.1$, log$_2$FC > ±0.58 thresholds. **G** TMM-normalized counts of log$_2$-transformed protein-coding exRNAs with highest variance across samples. Color scale indicates counts. Sample name: (M male, F female)_participantID_(pre; post)_(E, advanced; N, novice). **H** Reactome pathway analysis of protein-coding exRNAs enriched in pre-vs-post intervention exosomes. Color indicates $p$-value; circle size indicates exRNA count per pathway.

## Machine learning and feature-MEQ correlation

To identify features differentiating timepoints and experience levels across metabolomic, proteomic, fMRI, and RNA outcomes, we evaluated the data using Random Forest and XGBoost models. After normalization and dimensionality reduction, performance was assessed with tenfold cross-validation using F1 score, precision, recall, and area under the receiver operating characteristic (AUROC). Both models demonstrated robust classification across conditions, with AUC values ranging from 0.70 to 0.93. SHAP analyses revealed key contributors spanning metabolites, fMRI connectivity features, proteins, and non-coding RNAs (Fig. 8A, B).

Focusing first on timepoint classification, both models achieved strong discrimination between pre- and post-meditation states (XGBoost AUC = 0.86; Random Forest AUC = 0.90). Top XGBoost predictors included Gly-Gly-Phe, 3-indolebutyric acid, Gly-Lys, and connectivity between the anterior and salience networks, default mode–salience, and precuneus regions. Random Forest highlighted overlapping features alongside additional metabolites such as Androstenediol, S1P(d18:1), N6-methyladenosine, and purine derivatives. Many of these metabolites are linked to amino acid metabolism, neurotransmitter balance, and cellular stress signaling, while sphingolipids such as Androstenediol and S1P(d18:1) implicate lipid signaling and neuroendocrine function. Connectivity differences involving

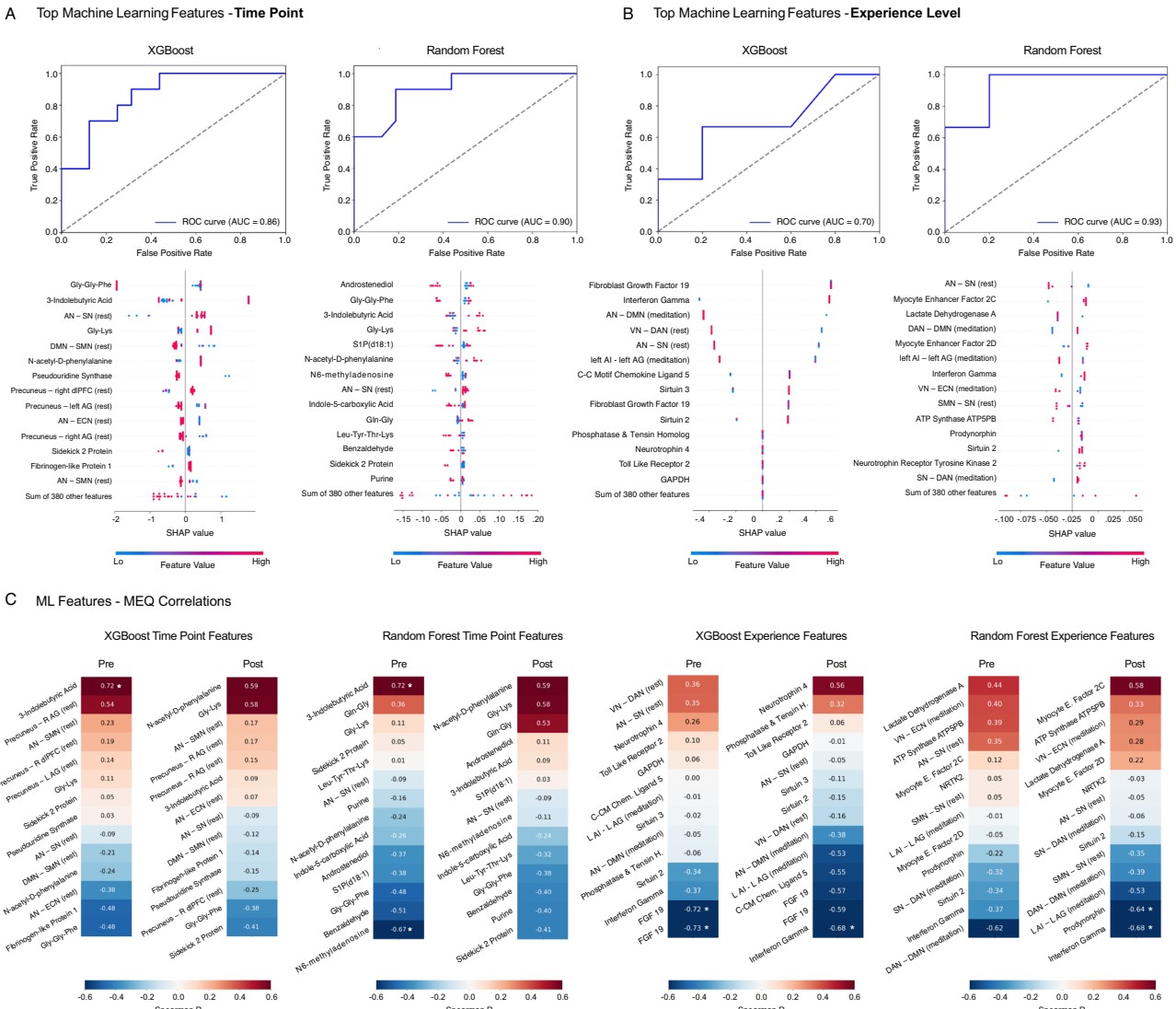

**Fig. 8 | Machine learning and MEQ-features correlations ($n = 20$ participants). A** ROC curves and SHAP plots for XGBoost and Random Forest models predicting pre/post classification: AUC = 0.86 (XGBoost) and 0.90 (Random Forest) indicate good classification performance. SHAP plots display the top contributing features ranked by impact on model output. **B** ROC curves and SHAP plots for XGBoost and Random Forest models predicting novice/advanced classification: AUC = 0.70 (XGBoost) and 0.93 (Random Forest). Abbreviations: R (right), L (left), AN (auditory network), SN (salience network), DMN (default mode network), SMN (somatomotor network), ECN (executive control network), VN (visual network), DAN (dorsal attention network), dlPFC (dorsolateral prefrontal cortex), AI (anterior insula), AG (angular gyrus), GAPDH (glyceraldehyde-3-phosphate dehydrogenase), ATP5PB (ATP synthase peripheral stalk membrane subunit B), FGF (fibroblast growth factor), NTRK2 (neurotrophin receptor tyrosine kinase 2). **C** Heatmaps of Spearman R correlations between Mystical Experience Questionnaire (MEQ) scores and the top machine learning features per model and time point, with * denoting FDR-adjusted statistical significance ($p_{FDR} < 0.05$).

salience, executive and default mode networks (SN, ECN, DMN, PC) align with prior literature that mediation can reorganize large-scale brain networks that govern salience[47], attention[48], interoception[49], and self-referential processing[47]. Taken in concert, timepoint differences identify shifts that integrate across neurofunctional and metabolic layers, reinforcing the concept that mediation acts as a systemic regulator of mind-body physiology.

To evaluate the influence of meditation experience, post–pre deltas were compared between novice and advanced practitioners. XGBoost achieved modest classification performance (AUC = 0.70), while Random Forest reached higher discrimination (AUC = 0.93). SHAP analysis indicated that immune- and stress-related proteins, including fibroblast growth factor 19, interferon-γ, C-C motif chemokine ligand 5, and Toll-like receptor 2, were major contributors in XGBoost, whereas Random Forest emphasized metabolic and mitochondrial features such as myocyte enhancer factors, lactate dehydrogenase A, ATP synthase ATP5PB, and

Sirtuin 2. Across both models, large-scale connectivity differences involving salience, executive, and default mode networks (e.g., AN–SN, VN–DMN, SN–DAN) emerged as consistent drivers of classification. These findings indicate that long-term meditation experience is associated with distinct molecular and network-level adaptations that support resilience and energy regulation.

Finally, to explore links between biological features and subjective outcomes, we correlated model-identified predictors with Mystical Experience Questionnaire (MEQ) scores. Several metabolites and connectivity measures showed nominal associations ($p < 0.05$), with a subset surviving FDR correction (asterisks, Fig. 8C). In the timepoint analysis, baseline 3-indolebutyric acid correlated positively with MEQ ($r = 0.72$, $q < 0.05$), whereas N6-methyladenosine correlated negatively ($r = -0.67$, $q < 0.05$). At the experience level, immune markers including FGF19 and interferon-γ were negatively associated with MEQ deltas ($r = -0.59$ to $-0.73$, $q < 0.05$), and reductions in network connectivity (DAN–DMN,

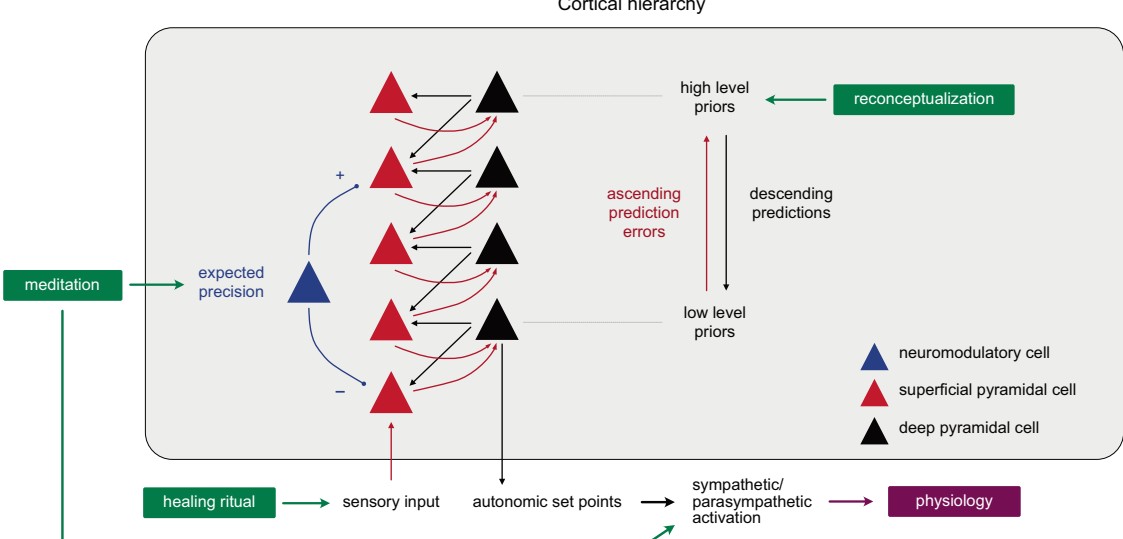

**Fig. 9 | Potential cortical implementation.** Hierarchical predictive coding scheme showing how the three mind-body techniques may synergistically facilitate a more flexible and adaptive predictive system: Reconceptualization remodels priors and hyper-priors; the healing ritual, acting as an open-label placebo, opens the system to new predictive avenues; and meditation weakens descending predictions to facilitate their replacement with new beliefs and present-centered sense data and activates sympathetic and parasympathetic autonomic responses. New predictions alter allostatic setpoints and autonomic regulation.

LAI–LAG) were inversely correlated with MEQ outcomes in experienced meditators ($r = -0.64$ to $-0.68$, $q < 0.05$). Together, these exploratory findings suggest that both peripheral metabolic and immune markers and central network dynamics are tied to reported mystical experience levels during meditation.

## Discussion

Our study shows how an intensive non-pharmacological mind-body intervention produced broad short-term neural and plasma-based molecular changes associated with enhanced neuroplasticity, metabolic reprogramming, and modulation of functional cell signaling pathways.

fMRI data showed that this meditation style functionally disrupts the default mode and salience networks (responsible for self-referential thought and allostatic regulation[32,50]) and cerebellum-prefrontal predictive processing circuits involved in integrating internal models with external sensory data[51]. This complements studies showing meditation-induced changes in DMN connectivity[32,52]. Mindfulness meditation has been linked to DMN deactivation[32,53] and stronger connectivity between DMN-Salience and DMN-Executive Control networks[47], likely reflecting an increased capacity to switch in/out of default mode dominance. Here, the main effect we observed was a meditation-driven decrease in intra-network DMN connectivity and a broad desynchronization of whole-brain connectivity. Meditation also reduced prefrontal-cerebellar connectivity both pre- and post-intervention, suggesting a state-dependent suppression of self-referential and evaluative processing consistent with focused interoceptive awareness, non-judgmental awareness, and a state that transcends the self— all features of the guided meditations. DMN-cerebellar connectivity alterations have been reported for depressive disorder and during previous meditation studies, albeit with opposite (higher connectivity) effects[54,55]. Our machine learning analyses further reinforced these findings by demonstrating that meditation-related shifts in fMRI connectivity were embedded within broader molecular changes across metabolomic, proteomic, and transcriptomic layers. The convergence of neural and peripheral predictors, including amino acid metabolites, lipid signaling molecules, immune regulators, and network-level connectivity, underscores that meditation engages a systemic mind-body axis rather than isolated pathways.

We can interpret these findings with a Bayesian brain framework, which posits that the brain probabilistically predicts incoming sensory information based on prior beliefs to allostatically regulate the body's energy needs, with experience and behavior largely shaped by the system's priors and predictive architecture[56]. Predictive brain regions project efferent copies to the subcortical nuclei that regulate metabolic, endocrine, immune, and other physiological setpoints[33,57] based on current and predicted energy needs. Thus, physiological and psychological health can be targeted by shifting the priors and functioning of the predictive system.

Each mind-body technique in the study may be thought of as acting on a different part of the predictive architecture (Fig. 9): Reconceptualization changes how participants believe their minds construct reality and affect their bodies, which is equivalent to reconfiguring the priors (beliefs about the causes of symptoms or sense data) and hyperpriors (beliefs about beliefs about the causes of sense data) that structure how interoceptive and exteroceptive sensory signals are interpreted and experienced. The open-label placebo-like healing ritual enacts a known health-promoting behavior (healing) that does not conform to rational norms, creating a mismatch between motor/affective behavior and the experience-driven self-fulfilling prediction of sense data, which opens the system to new predictive (sensory and allostatic) paths[30]. Finally, meditation, by focusing attention on the present, weakens the predictive processes themselves to produce a state of pure awareness that is free from the predicted self[50] in which priors are more easily replaced with present-centered sense data. The three techniques may therefore work synergistically to facilitate a more flexible and adaptive prediction system, accounting for the personal transformations anecdotally reported by participants and the downstream neuroplastic and molecular changes. The observed changes in blood biomarkers may have resulted from the endocrine, immune, and other regulatory changes evoked by the intervention's effects on these predictive allostatic mechanisms, as well as from meditation-induced effects on both sympathetic and parasympathetic branches of the autonomic nervous system[22], a finding common to other meditation studies[58], which may also account for the concomitant upregulation of inflammatory and anti-inflammatory proteins during the intervention.

MEQ-30 scores revealed that participants experienced mystical-type experiences during meditation, reflected in neural activity by strengthened connectivity between the left insula and the PCC—a hallmark sign of trance states—and diminished DMN connectivity and whole-brain modularity— consistent with previous meditation studies[32,59]. Elevated post-intervention plasma levels of SLITRK1 and NGFR and the inducement of greater glutamatergic dendritic growth by post-intervention plasma are both similar to

effects by serotoninergic entheogens, which some have compared to meditative states[60,61], and consistent with neuroplastic changes mediated by the BDNF pathway. While follow-up mechanistic studies are needed to establish causality, we provide converging evidence at multiple scales to suggest that the mind-body intervention produced neuroplastic changes mediated by these BDNF-associated proteins.

Beyond neural changes, we found that post-intervention plasma enhanced glycolysis in treated cells and contained higher levels of glycolysis-associated proteins ENO2 and LDHA. While the in vitro results may be interpreted as a Warburg effect on neuroblastoma cells, our proteomic data suggests that glycolysis became enhanced in participants. Notably, a similar phenomenon was observed in experienced Tibetan monks[24], for whom the enhanced glycolytic phenotype was associated with a cardioprotective plasma proteome, improved oxygen release, and decreased atherosclerosis. This underscores the complex interplay between neural and non-neural mechanisms in mind-body interventions and warrants further investigation. Since the brain's predictive architecture is evolutionarily geared towards regulating metabolic outlays, we speculate that the present-centered, sensory-driven brain state induced by meditation requires a more dynamic energetic profile made possible by glycolysis' fast response times.

The significant pre-to-post intervention increases in beta-endorphin and dynorphin point to the engagement of the endogenous opioid system without deception. We suggest that reconceptualization altered high-level priors about the mind's ability to influence the body, leading to new predictions that modulated endorphin levels. Since reconceptualization creates a steady belief state, this effect is likely sustained for longer than deception-based placebos. In a world in which up to 50% of American physicians regularly and knowingly prescribe placebo medications[62], interventions that activate a self-mediated, conscious, and non-deceptive pain relief mechanism carry great potential, especially for pain conditions without a well-described physical etiology. Future research should investigate which specific beliefs associated with health, disease, mind, and body sustainably optimize endogenous opioid modulation, as well as if and how meditation aids in initiating and sustaining these changes.

Our transcriptomic analyses reveal significant changes in circulating exRNAs implicated in neural activity, metabolic processes, and cellular signaling. Specifically, key upregulated transcripts such as *RIBC1* (*RIB43A* domain with coiled-coils 1) and *MIR455* (microRNA 455) mapped to pathways involved in neurotransmission and glycolysis. Pathway enrichment analysis highlighted upregulated Reactome pathways related to synaptic transmission and neurotransmitter cycling, consistent with the observed neural connectivity and neuroplasticity changes. Together, these results provide multi-level evidence that the mind-body intervention induced metabolic reprogramming, neuroplastic changes, and endogenous opioid modulation, potentially mediated by dynamic exRNA activity and glycolytic adaptation.

Interestingly, simultaneous activation of inflammatory and anti-inflammatory protein pathways is reminiscent of prior findings showing mind-body techniques can enhance resilience to environmental stressors via gene expression[63]. Cytokines like IL-6 are known to have both pro-and anti-inflammatory roles depending on the context acting as both acute-phase reactants during stress, but also stimulating IL-1 receptor antagonist and IL-10 in exercise and repair settings[64]. While inflammation is often viewed as a harmful response, increased cellular turnover and tissue remodeling could occur as part of an adaptive response to these interventions – another area worthy of further investigation.

While our study provides valuable insights into the broad effects of mind-body interventions, we acknowledge several important limitations. First, the uncontrolled observational design limits our ability to infer causality or disentangle the relative contributions of meditation, placebo, reconceptualization, and other factors such as expectations, diet, and relaxation or distance from routine stressors. Future studies should implement more rigorous, controlled designs to test how these elements interact and whether the non-predictive meditation state facilitates the replacement of maladaptive priors during reconceptualization. Additionally, the small

sample size and partial reliance on experienced meditators may limit the generalizability of our findings, and future research should include larger, more diverse cohorts. Additional mechanistic studies should investigate the molecular targets implicated by this study to causally link the neuroplastic and phenomenological changes to the molecular pathways implicated by our work.

Specific methodological limitations include the following. Although data on meditation experience and practice frequency were collected for retreat-based meditations within the study cohort, equivalent information regarding other forms of meditation practice—apart from the binary indication of their presence or absence—was not obtained. Potential circadian and metabolic confounds were introduced by variable blood collection times (up to 8-h range on Day 8), pre-collection fasting durations, and time elapsed after the meditation intervention ended (up to 48 h for blood collections and 24 h for fMRI acquisition) between participants. Dietary factors and fasting may also have introduced confounds in proteomic and metabolomic measures since no standardized diet was implemented and fasting times beyond 30-min pre-blood collection were not controlled during the intervention. The short duration of the resting state scan can limit our ability to attribute connectivity-derived features solely to the experimental variables[64,65]. The open eyes paradigm during fMRI BOLD acquisition chosen to reduce the risk of drowsiness likely introduced potential visual confounds. Finally, denoising with ICA-AROMA + WM/CSF regression is limited in removing physiological noise in regions with strong cardiac or respiratory activity, which may have resulted in additional confounds.

## Conclusions
This observational study is the first to investigate the joint effects of three mind-body interventions–meditation, reconceptualization, and open-label placebo healing rituals–on neural activity and plasma physiology. Our findings show these techniques may act synergistically to produce mystical-type experiences, enhance neuroplasticity, reprogram metabolic pathways, and modulate endogenous opioids, highlighting the potential of mind-body interventions to effect profound changes in neural activity and physiology.

## Methods
### Participant recruitment
The study was advertised to all 1444 registered retreat attendees via email invitation, of which 561 expressed interest by answering an online questionnaire to determine eligibility (Fig. 1D). Inclusion criteria included being an English speaker; being at least 21 years of age; being in good general health; being able to provide blood samples before, during, and after the retreat; and agreeing to undergo fMRI neuroimaging before and after the retreat. Exclusion criteria included inability to consent, inability to follow or comply with study procedures, current use of psychoactive medications, contraindications for phlebotomy, and contraindications for MRI (pregnancy, history of seizures, electronic or ferromagnetic medical implants or devices, claustrophobia). Of 65 eligible participants, 36 were randomly selected to participate and provided written informed consent. Of 27 who consented, five were dropped due to scheduling conflicts; one participant was dropped for not complying with MRI facility masking requirements; and one participant failed MRI screening due to a heart stent. The study size was determined by the scanning time available at the fMRI facility (1 MRI scanner for 2 days pre- and post-intervention). Our study included male ($n = 6$) and female ($n = 14$) participants (gender identity), and similar findings are reported for both genders. Of the 6 male participants, 2 had been practicing the meditations carried out in the workshop daily for more than 1 year, 3 had been practicing the meditations daily for less than 1 year (one of whom practiced a different form of meditation), and 1 participant had not previously practiced the meditations before attending the retreat, though he had a different meditation practice. Of the 14 female participants, 8 had been practicing the retreat meditations for over 1 year (most being a daily practice), and 1 of the

8 also practiced a different form of meditation. Six of the 8 females had been practicing the meditations for less than 1 year, with only 3 reporting an almost daily practice, and 4 of the 8 reported having a different meditation practice. One of the 8 females had not previously practiced the meditations before attending the retreat, though she had a different meditation practice. The retreat began at 5 pm on Day 1 and ended at 1:30 pm on Day 7 and was conducted in April 2022 at the Manchester Grand Hyatt (San Diego, CA, USA). The diet was provided by the hotel. There were no adverse events.

The study was conducted in accordance with the Declaration of Helsinki principles and all relevant ethical regulations. Experimental protocols were approved by the Western Institutional Review Board (WIRB; now WCG-IRB; Protocol MED02#20211477) and registered on clinicaltrials.gov (NCT 06615531). Written informed consent was obtained from all participants prior to study inclusion, and clinical records are housed at VitaMed Research (Palm Desert, CA) as mandated by federal laws. All ethical regulations relevant to human research participants were followed.

### Functional magnetic resonance imaging (fMRI)

**Acquisition**. fMRI was performed off-site at the UCSD Center for Functional MRI pre-intervention (Day 0: 8 am–8 pm and Day 1: 8 am–4 pm) and post-intervention (Days 8 and 9: 8 am–6 pm). Participants were positioned in the MRI scanner (Siemens Prisma 3T with standard 32-channel head coil) with a respiratory transducer placed around the chest, a pulse oximeter placed on the left index finger, and MRI-safe headphones. A structural scan, two functional blood oxygen level dependent (BOLD) scans (resting state and meditation), and a diffusion tensor imaging (DTI) scan (not described here) were acquired.

**Structural scan**. Participants were instructed to "not move and keep eyes closed." High resolution structural images were acquired with a T1-weighted magnetization-prepared rapid gradient-echo (MPRAGE) sequence with TR = 2400 ms, TE = 2.22 ms, TI = 1000 ms, flip angle = 8°, FOV = 224 mm, voxel size = 0.7 mm isotropic, 320 slices, slice thickness 0.80 mm, saggital orientation, bandwidth = 210 Hz/Px, and acquisition time = 7 min. 40 s.

**Functional scans**. During the 5-min BOLD resting state scan, participants were instructed to "not move, keep eyes open, stay awake, and think about whatever you want, but do not meditate." During the subsequent 15-min BOLD meditation scan, an auditory recording of a guided meditation from the retreat was played and participants were instructed to "not move, listen to the guided meditation soundtrack, and meditate as suggested by the audio while keeping your eyes open." An eyes open paradigm was chosen to reduce the risk of drowsiness or sleep during the meditation run, which was deemed a potentially greater risk than the presence of visual confounds to accurately comparing active meditation with passive rest[66,67]. One rest and one meditation run were collected per scan. The audio consisted of continuous expansive atmospheric music with an intermittent ethereal voice repeatedly instructing listeners to "tune into nothingness", "no time", "nowhere", and "love", combining loving-kindness meditation with focused awareness of the experiencing self and its potential dissolution into pure awareness. BOLD images were obtained with a gradient-recalled echo-planar imaging (EPI) sequence with TR = 800 ms, TE = 37 ms, flip angle = 52°, FOV = 208 mm, voxel size = 2 mm isotropic, 2 mm slice thickness, 72 slices, number of volumes = 1200, matrix size = 104 × 90, bandwidth = 2290 Hz/Px, and acquisition time = 5 min. (rest) and 15 min. (meditation).

**Post-scan questionnaire**. Immediately after each scanning session, participants were asked to assess their experience meditating by completing the Mystical Experience Questionnaire (MEQ-30)[68].

**fMRI preprocessing and denoising**. Anatomical and functional data were preprocessed using fMRIPrep v21.0.2[69]. B0-field maps were estimated with Topup[70]. Anatomical T1-weighted (T1w) images were corrected for intensity non-uniformity using ANTS 2.3.3[71], skull-stripped, segmented using Fast (FSL 6.0.5)[72], and normalized to standard space (MNI152Nlin2009cAsym) with nonlinear registration (ANTs 2.3.3). Brain surfaces were reconstructed using recon-all (FreeSurfer 6.0.1)[73]. Due to a corrupted T1w image for participant 4023, co-registration failed, and 4023 was excluded from fMRI analysis.

BOLD head motion parameters were estimated using MCFLIRT (FSL 6.0.5)[74]. Functional runs were co-registered to anatomical T1w references using boundary-based registration (FreeSurfer) and resampled to standard space. Confounding time-series were calculated for CSF/WM region-wise global signals, and motion artifacts were identified using independent component analysis (ICA-AROMA)[75]. Non-steady state volumes were removed and spatial smoothing with an isotropic Gaussian kernel of 6 mm FWHM (full-width half-maximum) was applied. Denoising using Nilearn 0.9.2[76] was performed by detrending, standardizing, and bandpass filtering (0.01–0.1 Hz) time series data, and by regressing ICA-AROMA motion artifacts and mean CSF + WM signals[75]. Physiological noise was addressed via ICA-AROMA + CSF/WM regressions rather than with physiological recordings. Additional denoising for whole-brain network analyses involved regressing global signal and excluding BOLD runs with mean FD > 0.3 mm to obtain functional connectivity distributions mean-centered around zero, which were confirmed by visual inspection.

**Functional connectivity analysis**. Functional connectivity analyses were performed on seven canonical resting state networks (RSNs), eight a priori defined regions of interest (ROIs), and whole-brain networks (Nilearn 0.9.2) (Brain Connectivity Toolbox[77]). To understand how this meditation style influences large-scale neural dynamics, we examined functional connectivity and whole-brain network measures, capturing the integration, segregation, and reorganization of brain networks.

**Resting state networks and regions of interest**. RSNs examined included the default mode network (DMN), dorsal attention network (DAN), executive control network (ECN), salience network (SN), sensorimotor network (SMN), visual network (VN), and auditory network (AN). The Montreal Neurological Institute (MNI) coordinates for the 36 regions that comprise them were extracted from the Raichle (2011) atlas[78] (Supplementary Table 3). Mean denoised BOLD time series were extracted from 10 mm-radius (523-voxel) spheres centered on the MNI coordinates. Between-region Pearson correlations were calculated and z-transformed to obtain a measure of connectivity strength. Within-network connectivity was calculated as the mean connectivity between each unique pair of ROIs within a given network, and between-network connectivity was calculated as mean connectivity between all ROIs from two networks, with each ROI pair containing one region per network. Eight a priori ROIs were additionally selected for a hypothesis-driven analysis based on task-induced and connectivity changes in other meditation studies[79–82]: medial prefrontal cortex (mPFC), right and left dorsolateral prefrontal cortices (r/l dlPFC), right and left insular cortices (r/l IC), right and left angular gyri (r/l AG), and central precuneus, with MNI coordinates extracted from the DiFuMo atlas[83] (Supplementary Table 4). Mean denoised BOLD time series per region were extracted from 10 mm-radius (523-voxel) spheres centered on the MNI coordinates, and between-region Pearson correlations were calculated.

**Whole-cortex/brain networks**. Networks were constructed from the 25%-thresholded 48-region Harvard-Oxford 2 mm cortical atlas[37] and from the 264-region whole-brain Power (2011) atlas[38] by extracting mean denoised BOLD time series per region per parcellation and calculating between-region Fisher z-transformed correlations for all possible pairs. We constructed weighted undirected graphs and computed modularity, global efficiency, and characteristic path length per network. Functional connectivity value distributions were visually inspected, and one outlier run not mean-centered around zero was excluded from Power atlas

analysis. Network measures were recalculated while excluding runs with mean framewise displacement > 0.3 mm to check if any significant effects were due to motion artifacts.

**Statistics.** We investigated scanner head motion per session and condition with a 2 (pre/post) × 2 (meditation/rest) repeated measures ANOVA on mean and maximum framewise displacement per BOLD run with post-hoc effects confirmed with Wilcoxon signed rank tests. MEQ score differences were compared between sessions with Wilcoxon signed rank tests ($n = 18$, two subjects missing data).

After confirming normality and heteroskedasticity, paired $t$-tests compared RSNs, their component regions, a priori ROIs, and whole-cortex/brain networks pre- and post-intervention (separately for rest and meditation scans), plus rest and meditation (separately for pre- and post-scans) to investigate effects of time and of meditation, respectively. On rest vs. meditation comparisons, the last 5 min of the 15-min meditation scan were used to compare equal-length scans. $P$-values were corrected for multiple comparisons using Bonferroni–Holm family-wise error correction and Benjamini–Hochberg false discovery rate when no results reached $p_{FWE} < 0.05$.

**Anatomical data.** Preprocessed skull-stripped segmented T1w data was resampled to the MNI fsaverage template using mris_preproc (Free-Surfer 7.3.2) and cortical surface was inflated to an average spherical surface. Hemispheres were automatically parcellated into regions of interest, and vertex-wise volumetric and cortical thickness calculations were performed. Images were spatially smoothed with a 5 mm FWHM Gaussian kernel. Pre-vs-post differences were assessed by repeated measures ANOVA, and group differences between novice and advanced participants at each timepoint were assessed by fitting a general linear model using mri_glmfit-sim. Total intracranial volume and age were included as covariates, and results were cluster corrected with a cluster-wise $p$-threshold = 0.01.

**Human plasma collection**
Human plasma was collected and processed, as described in ref. 84, via venous puncture by registered nurses, physicians, and phlebotomists, at pre-intervention either off-site ("Day -1", 2 days pre-retreat, 2–4 pm) or on-site ("Day 0", 1 day pre-retreat, 11 am–4 pm) and post-intervention ("Day 8", on-site, 9–5 pm). Participants were offered the same food options for all breakfast, lunch, and snacks throughout the week, although food choices were not monitored. All participants were required to fast for at least 30 min prior to blood collection. Blood was collected in EDTA-coated tubes (BD, Franklin Lakes, NJ) and kept at 4 °C (wet ice) for less than 30 min. Plasma was isolated by centrifugation at 3000 RPM for 15 min in an E8 Touch tabletop centrifuge (LW Scientific, Lawrenceville, GA), aliquoted into 1.5 mL Eppendorf tubes, and immediately frozen on dry ice. At the end of the retreat, samples were shipped to UCSD and stored at −80 °C.

**Plasma nanoparticle tracking analysis**
Plasma Nanoparticle Tracking Analysis (NTA) measured circulating plasma nanoparticle size and concentration with a NanoSight NS300 instrument (Malvern Instruments Ltd., UK). Samples were thawed on ice and diluted in PBS (100×) to prevent aggregation. Measurements were performed at room temperature under continuous video recording (16 × 30-s. acquisitions/sample) and 532 nm laser illumination. Mean hydrodynamic diameter, size distribution, and particle concentration were obtained per sample with NTA 3.3 software.

**ELISA**
To investigate plasma concentrations of oxytocin, beta-endorphin, dynorphin, anandamide, cocaine, and amphetamine-regulated transcript (CART), c-reactive protein (CRP), and neuropeptide Y (NPY), pre- and post-intervention plasma samples were tested with commercially available

ELISA kits listed in Supplementary Table 12 following manufacturers' instructions. Absorbance was measured with a Tecan Spark 10 M microplate reader, and concentrations were calculated by interpolating absorbance values against the standard curve, previously fit with a 3-parameter logistic curve. Pre- and post-intervention concentrations were compared with Wilcoxon signed rank tests.

**Plasma proteomics**
To investigate the intervention's effects on the plasma proteome, 7596 proteins were quantified with the SomaScan Assay v4.1 (SomaLogic, Boulder, CO, USA).

**SomaScan assay.** Samples were thawed on ice, diluted in SomaLogic plasma diluent, and loaded onto SomaScan 96-well plates containing capture SOMAmer reagents for 7596 unique human protein targets. Plates were incubated for protein capture, unbound material was washed away, and biotinylated capture antibodies were added to hybridize with SOMAmer-bound proteins. Streptavidin-conjugated Cy3 dye was added to label the antibodies, plates were washed to remove unbound dye, and protein-bound SOMAmer reagents were eluted and hybridized to custom DNA microarrays containing complementary sequences to each SOMAmer. Cy3 fluorescence intensity in relative fluorescence units (RFU) measured protein abundance. Hybridization control normalization removed sample variance between microarrays and scanners; median signal normalization removed within-plate inter-sample differences; and calibration normalization removed variance across assay runs. Performance and quality were monitored with blank wells, technical replicates, and spiked protein controls, and precision was assessed by calculating coefficients of variation for all measurements. Background subtraction RFU calculations were performed with SomaLogic Discovery Server software.

**Data processing.** RFU values were normalized, log10-transformed, and auto-scaled (mean-centered and divided by each variable's standard deviation), and outliers above two SDs were flagged. Foldchange and paired $t$-tests tested pre/post-intervention differences, and hierarchical clustering and k-means clustering revealed correlated variation expression patterns. Functional enrichment analysis (Enrichr-KG) on differentially expressed proteins (pre/post $p < 0.01$), with all SOMAmer targets as background, assessed enriched biological processes, molecular pathways, and cellular components. Enriched pathways were considered significant at $p_{FDR} < 0.05$. Protein-protein interaction subnetworks of significant proteins ($p < 0.05$ or |fold change| $\geq 0.5$) were created in Cytoscape with $\leq 30$ interactor nodes and heat diffusion using the STRING database[85].

**Plasma metabolomics**
640 metabolites in whole plasma were detected via liquid chromatography-mass spectrometry analysis using the widely-targeted Metware platform (Metware Biotechnology, Woburn, MA, USA).

**Metabolite detection.** Plasma samples were extracted with a 1:4 ACN:methanol solution and centrifuged at 12,000 RPM (10 min, 4 °C) and again at 12,000 RPM (3 min, 4 °C) after 30 min. at −20 °C. Supernatants were analyzed by ultra-HPLC (UPLC) (ExionLC 2.0, Sciex, Framingham, MA, USA) on a UPLC column with a gradient elution system for metabolite separation (UPLC conditions listed in Supplementary Table 13). Eluted metabolites were detected based on mass and fragmentation patterns by Tandem Mass Spectrometry (MS/MS) using a quadrupole-time of flight mass spectrometer (QTRAP®6500+, Sciex) with an electrospray ionization (ESI) Turbo Ion-Spray interface, operating in positive and negative ion mode. Untargeted qualitative metabolite identification was performed by matching ion features to references from Metware, HMDB[86], METLIN[87], and KEGG[88] databases. Identified metabolites were then quantified using triple quadrupole mass

spectrometry with multiple reaction monitoring (mass spectrum conditions in Supplementary Table 14). Mass spectrometry chromatographic peaks were corrected with MultiQuant software (Sciex). Reproducibility, cross-contamination, and inter-sample quality were controlled with control samples, internal standard peaks from blank samples, and internal standards, respectively.

**Metabolite analysis.** Quantified metabolite features were variance-filtered via inter-quartile range (max 25% filtered out), normalized, and log10-transformed. Outliers above 2 standard deviations were flagged. Dietary factors and fasting may have introduced confounds in proteomic and metabolomic measures, as no standardized diet was implemented and fasting times beyond 30 min pre-blood collection were not controlled. Exogenous compounds annotated as dietary or drug-related by HMDB were monitored using the MetaboAnalyst pipeline and were flagged or removed if observed among the top 25 features. PCA, hierarchical cluster analysis (HCA), partial least squares discriminant analysis (PLS-DA), and FDR-corrected pre/post paired $t$-tests (Scikit-learn[89]) identified differentially expressed metabolites. Correlation heatmaps were generated using Pearson $r$ distance measures. Metabolic pathway analysis comparing pre- to post-samples was performed with MetaboAnalyst v5.0 based on the KEGG database. Pathway enrichment was first assessed using MetaboAnalyst, and false discovery rate (FDR) correction was applied to control for multiple comparisons. Significant pathways ($p_{FDR} < 0.05$) were then selected for follow-up analysis. For these pathways, we examined the distribution of normalized feature values and report the corresponding $p$-values for group comparisons after normalization.

**Real time cellular metabolic analysis**
To investigate the effects of human plasma on cellular metabolism, we performed Seahorse XF mitochondrial stress, glycolytic rate, and ATP production assays on BE(2)M17 cells (ATCC), a human neuroblastoma cell line, exposed to plasma following[90,91] methods.

**Cell culture and plasma treatment.** BE(2)M17 cells were cultured in complete DMEM/F12 medium supplemented with 10% FBS and 1% penicillin/streptomycin under standard conditions (37 °C, 5% $CO_2$). Cells were harvested, washed, and seeded onto microplates pre-coated with poly-L-lysine at a 40,000 cells/well density and incubated overnight. Cells were resuspended in XF assay medium and treated with 1% plasma for 1 h at 37 °C, with 4 technical replicate wells per plasma sample, and stained with DAPI for post-assay cell counting.

**Seahorse XF assays.** Seahorse XF96 Analyzer (Agilent, Santa Clara, CA, USA) was calibrated according to manufacturer's instructions, and the following injection protocols were implemented: Mitochondrial Stress Test (kit #103015-100): Oligomycin (1.5 μM), FCCP (2.0 μM), and Rotenone/Antimycin A (0.5 μM) were sequentially injected to measure basal respiration, proton leak, maximal respiration, and spare respiratory capacity, with oxygen consumption rate (OCR) monitoring. Glycolytic Rate Test (kit #103344-100): Rotenone/antimycin A (0.5 μM) and 2-deoxyglucose (50 mM) were injected to assess basal glycolytic rate, glycolytic reserve, and non-glycolytic acidification, monitored by extracellular acidification rate (ECAR). ATP Production Rate Test (kit #103592-100): Injections of oligomycin (1.5 μM) and rotenone/antimycin A (0.5 μM) enabled real-time calculation of ATP production rates from OCR and ECAR.

**Data analysis and statistics.** We used Seahorse XF software (v.2.6) to analyze raw OCR and ECAR data, normalizing results by cell counts, and compared metabolic parameters between control, pre-, and post-intervention groups with paired $t$-tests ($p < 0.05$ considered significant). We assessed cell viability pre- and post-treatment and included four non-cellular blank well controls per plate to ensure assay quality.

**Neurite differentiation**
To investigate the effects of human plasma on neurite growth, PC12 neuroendocrine cells were differentiated to their neuronal phenotype, treated with plasma, and live-imaged following published methods[92].

**Cell culture and differentiation.** PC12 cells (ATCC, Manassas, VA, USA) were cultured in RPMI 1640 medium supplemented with 10% horse serum, 5% FBS, and 1% penicillin/streptomycin under standard conditions (37 °C, 5% $CO_2$). On Day 0, differentiation was induced by plating cells onto poly-D-lysine-coated plates at a $1.0 \times 10^4$ cells/cm$^2$ density and culturing them with Opti-MEM medium supplemented with 0.5% FBS, 1% penicillin/streptomycin, 50 ng/mL nerve growth factor (NGF), and either 1% human plasma pooled from pre- and post-intervention or no plasma (control cells) with 2 technical replicate wells per condition. Medium was replaced every 48 h. Media was supplemented with 1% Culture One from Day 2 onwards to support neuronal differentiation.

**Live-cell imaging and neurite analysis.** Differentiated cells were live-imaged every 24 h for 10 days at 20× in a live cell imaging microscope (Keyence BZ-X700) equipped with a phase contrast objective and an incubation chamber at 37 °C, 5% $CO_2$, and controlled humidity. Images were acquired under consistent illumination across groups and time points, and background subtraction and image thresholding were applied to enhance neurite visualization. The single longest neurite per cell, for cells for which it was longer than the cell body diameter, was manually traced in FIJI software (v1.0). Neurite lengths were compared with two-tailed independent samples $t$-tests between plasma-treated and control groups at each time point. Treatment wells remained blinded until completion to avoid bias.

**Exosome-specific small RNA transcriptomics**
**Exosome isolation.** Exosomes were isolated from plasma samples (300–500 μl) by centrifugation at $300 \times g$ for 10 min at 4 °C to remove cell debris. Supernatants were transferred to microcentrifuge tubes, filled with 1× PBS, balanced for mass, placed in a pre-cooled Beckman Coulter Type 70.1 rotor, and centrifuged at $10,000 \times g$ for 30 min at 4 °C with maximum acceleration and gradual deceleration. Supernatants were transferred to new tubes, exosome pellets were resuspended in 100 μl 1× PBS, and another round of ultracentrifugation at $100,000 \times g$ for 2 h. at 4 °C was performed. Final exosome pellets were resuspended in 150 μl 1× PBS, analyzed by nanoparticle tracking using the same protocol as whole plasma, and stored at −80 °C.

**Exosome RNA sequencing.** Total RNAs were purified from exosome pellets using Direct Zol RNA mini kit (Zymo Research). RNAs were eluted in 20 μl of RNAse-free water and concentrated to 5 μl using SpeedVac Vacuum Concentrator (Thermo Fisher Scientific). 44 to 523 ng of exosome-specific extracellular small RNAs (exRNAs) were obtained from 28 human plasma samples. exRNA quantity and integrity were determined using the NanoDrop ND-1000 spectrophotometer (Thermo Fisher Scientific) and the Bioanalyzer 2100 (Agilent, Santa Clara, CA, USA), respectively. Small exRNA libraries were generated using NEXTFlex Small RNA-seq Kit v4 with UDIs (Revvity Inc, Waltham, MA, USA) according to modified manufacturer's protocol to account for low exRNA concentrations in exosome samples. The strategy for small RNA libraries aimed at a range of 16–120 nucleotide (nt)-long transcripts for accurate identification and quantification of exosome-specific extracellular microRNAs, ncRNAs, and protein-coding RNAs. NEXTFLEX® 3′ Adenylated Adapter v4 was ligated at 25 °C for 1 h. Following 3′ Adenylated Adapter inactivation, NEXTFLEX 5′ Adapter v4 was ligated at 20 °C for 1 h. Reverse Transcription-First Strand Synthesis from 5′ and 3′ NEXTFLEX® Adapter Ligated RNA templates was conducted at 50 °C for 1 h, followed by inactivation at 90 °C for 5 min. First Strand Synthesis products were purified using NEXTFLEX Cleanup

Beads by two washes in 80% ethanol. RNA templates were combined with NEXTFLEX UDI Barcoded Primer Mixes v4 diluted 1:4 and amplified by PCR for 24 cycles. PCR products size selection and cleanup were done using NEXTFLEX Cleanup Beads. Small exRNA libraries were eluted from beads in 15 µl and concentrations were determined using Qubit dsDNA HS Assay (Life Technologies). DNA fragment profile in small exRNA libraries was analyzed on the Bioanalyzer 2100. RNA-seq was conducted on the NovaSeq × Plus Short-Read Sequencer (Illumina, San Diego, CA) at the Genomics Research and Technology Hub, University of California, Irvine.

**RNA-seq data processing.** Sequencing read data were converted to FASTQ format for bioinformatic processing using Lonestar6 supercomputer at the Texas Advanced Computing Center and bioinformatically processed according to guidelines developed for NEXTFLEX Small RNA-Seq (Revvity Inc., Waltham, MA, USA). In brief, flanking forward (TGG AAT TCT CGG GTG CCA AGG) and reverse (AGA TCG GAA GAG CGT CGT GTA GGG AAA GA) adapter sequences were trimmed prior to alignment with Cutadapt[93]. Paired reads above 16 nucleotides with quality scores above 20 were mapped to the human genome primary assembly (Release 47, GRCh38.p14). Reads-to-genome alignment was conducted using STAR aligner[94]. Mapping quality was assessed using MultiQC tool[95].

**Detection of differential exosomal RNAs.** The differential analysis of sequence read count data was done using a generalization of a paired *t*-test using EdgeR software[96–98] in which exosome RNA pools in pre- and post-intervention samples per participant were compared separately and the baseline differences between participants were subtracted. Transcripts with fewer than 10 counts were removed from analysis. Trimmed Mean of *M*-values (TMM) normalization[99,100] was applied to account for compositional difference between samples. exRNA were annotated using org.Hs.eg.db[101] and miRbase[102] databases. Sequences corresponding to non-coding RNAs (ncRNAs) and exhibiting the highest variance across all samples were analyzed by PCA of TMM-normalized counts using PCAtools Bioconductor package. Functional analysis of differential exRNA was done using ClusterProfiler[103] and the Reactome Pathways Knowledgebase[46]. A paired design in the negative binomial (NB) generalized linear model (GLM) detected the effect of the intervention on the composition of exRNAs from paired post-intervention ($n = 14$) vs pre-intervention ($n = 14$) samples, while adjusting for differences between participants. Participants' experience level (9 advanced, 5 novices), gender (8 females, 6 males), and age were included as additional covariates.

**Machine learning and MEQ-features correlation analyses**
We applied machine learning to identify the most biologically relevant features across time point (pre/post) and experience level (novice/advanced) datasets. Each dataset was preprocessed (log-transformed and auto-centered) and missing data was imputed to ensure feature scaling, normalization consistency, and data integrity and comparability across modalities[104]. Post-preprocessing, ELISA, metabolomics, transcriptomics, and proteomics datasets were concatenated into a single feature matrix used as input for an eXtreme Gradient Boosting (XGBoost)[105] classifier chosen for its ability to handle high-dimensional data. To obtain a unified impact metric per feature, the dataset was passed through a Shapley Additive exPlanations (SHAP)[106] explainer model. Spearman correlations were then calculated between MEQ scores and the top 14 features per model and corrected for multiple comparisons to explore whether the top molecular changes were linked to subjective outcomes.

**Statistics and reproducibility**
Given the $n = 20$ sample size ($n = 19$ for fMRI), nonparametric Wilcoxon signed rank statistics were performed throughout when comparing paired samples, except when $n \geq 30$ (e.g., protein targets) or when normality and homoskedasticity were ascertained, in which case two-tailed paired *t*-tests were employed. Given that proteomic and metabolomic analyses were hypothesis-free and data driven, False Discovery Rate (FDR) correction was applied using the Benjamini–Hochberg procedure, and reported *p*-values are labelled as FDR-adjusted, with nominal *p*-values included for transparency. Seahorse XF plates were prepared with 4 technical replicates (wells) per participant and time point. PC12 assay plates were prepared with pooled plasma ($n = 20$) with 2 technical replicates (wells) per treatment (pre-plasma/post-plasma/no treatment).

**Ethics and inclusion statement**
Data collection and analysis was performed locally in San Diego, CA, USA and included local researchers. Roles and responsibilities were agreed amongst collaborators ahead of the research, and clinical practice and biospecimen handling trainings were provided to all researchers. The study was approved by a local research ethics committee and did not result in personal risk to study participants. Researchers who handled biospecimens during collection and analysis wore adequate personal protective equipment. fMRI researchers and participants were thoroughly screened for MRI safety before being allowed inside the fMRI magnet room.

## Data availability
Values for all data points in plots and reported means are available (Supplementary Data 1). RNA-seq data is available from GEO repository (accession #GSE291700). Proteomics data is available from PRIDE repository (dataset PAD000010). Any other data is available from the corresponding author on reasonable request.

## Code availability
Analytic code can be obtained from the corresponding author on reasonable request.

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

## Acknowledgements
Support for this research was provided by the InnerScience Research Fund (H.H.P.) and a Research Career Scientist Award from the Veterans Administration (BX005229 to H.H.P.).

## Author contributions
A.J.D. contributed to FMRI design, data collection, and analysis, PC12 assay, ELISA assays, Seahorse assays, NanoSight measurements, exosome isolation, proteomics and metabolomics analyses, figure creation, and manuscript writing. S.S. contributed to manuscript writing, figure creation, and metabolomics and machine learning analyses. J.P.Z.H. contributed to Seahorse and NanoSight analyses. R.C. contributed to sample collection and processing and manuscript edits. J.M.S. contributed to sample collection and processing. J.A.B. contributed to data collection and manuscript edits. L.C. contributed to FMRI design, data collection, and interpretation. J.D. contributed to guiding the mind-body retreat intervention. J.M. contributed to participant recruitment, IRB, and data collection. N.E.A.S. contributed to PC12 neurite analysis. M.L. contributed to dataset curation and analysis. N.F. contributed to PC12 and ELISA assays. S.D. contributed to sample collection and processing. A.V.C. contributed to RNA-seq, bioinformatic analysis, and transcriptomics figure creation. L.C.M. contributed to analysis. N.R. contributed to methodological consultation and manuscript edits. M.A.P. contributed to study design, execution, and manuscript edits. T.M.B. contributed to study design and execution. H.H.P. contributed to study design, execution, manuscript edits, and funding.

## Competing interests
The authors declare the following competing interest: J.D. is employed by Encephalon, Inc., which offers the meditation retreats. All other authors declare no competing interests.
