## [Transparent Peer Review file · Communications Biology]

Neural and Molecular Changes During a Mind-Body Reconceptualization, Meditation, and Open Label Placebo Healing Intervention

Corresponding Author: Dr Hemal Patel

Version 0:

Decision Letter:

**** Please ensure you delete the link to your author homepage in this email if you wish to forward it to your coauthors ****

Dear Dr Patel,

We would like to sincerely apologize for the delay in reaching a decision your manuscript entitled "Neural and Molecular Changes During a Mind-Body Reconceptualization, Meditation, and Open Label Placebo Healing Intervention." Unfortunately, one of the reviewers had dropped out during the review process which led to unexpected delays. We acknowledge that we did not meet our expected standards.

Your manuscript has now been seen by 2 referees, whose comments are appended below. You will see from their comments below that while they find your work of considerable interest, some important points are raised. We are interested in the possibility of publishing your study in Communications Biology, but would like to consider your response to these concerns in the form of a revised manuscript before we make a final decision on publication.

We therefore invite you to revise and resubmit your manuscript, taking into account the points raised. In particular, please provide more details in the methods and results, as both reviewers note missing information that needs clarification. Please note that we do not impose word limits, and all methods should be described in sufficient detail to allow for reproducibility.

Please highlight all changes in the manuscript text file.

We are committed to providing a fair and constructive peer-review process. Do not hesitate to contact us if you wish to discuss the revision in more detail or if there are specific requests from the reviewers that you believe are technically impossible or unlikely to yield a meaningful outcome.

At the same time, we ask that you ensure your manuscript complies with our editorial policies. Specifically:

For all graphs depicting a single point value (e.g., mean) with error bars, you must add individual data points or convert the graph to a boxplot or dot-plot to show data distribution.

It's mandatory to provide access to the numerical source data for graphs and charts either through a repository or by providing the data in a Supplementary Data file (in excel format).

All blots/gels must be accompanied by size markers in every figure panel. Uncropped and unedited blot/gel images must be included as Supplementary Figure(s) in the Supplementary Information pdf.

Please ensure that you have complied with the data deposition policies at the Nature Portfolio, please see here.

Please ensure that you have complied with our policies on research involving animals and humans, see here

Please follow the ARRIVE guidelines for reporting animal experiments. Please fully complete an [ARRIVE checklist](https://arriveguidelines.org/sites/arrive/files/documents/Author%20Checklist%20-%20Full.pdf) including both the essential and recommended set of items (adding information to the manuscript where needed) and upload this with your revised manuscript.

Please also see [our revision checklist](https://www.nature.com/documents/CommsBio-file-checklist-revision.pdf) for guidance on formatting the manuscript and complying with our policies. A comprehensive guide to our formatting requirements for final submissions is also available for your reference [here](https://www.nature.com/documents/commsj-life-style-formatting-guide-accept.pdf).

Please use the following link to submit your revised manuscript, point-by-point response to the referees' comments (which should be in a separate document to the cover letter) and any additional files:

Link Redacted

When submitting the revised version of your manuscript, please pay close attention to our [Digital Image Integrity Guidelines](https://www.nature.com/commsbio/editorial-policies/image-integrity).

We would like to receive your revision within 2 months, but appreciate that every situation is unique. We look forward to receiving your revised manuscript when it is ready, and will not enforce a hard deadline on this revision.

Please do not hesitate to contact me if you have any questions or would like to discuss these revisions further. We look forward to seeing the revised manuscript and thank you for the opportunity to review your work.

Best regards,

Jasmine Pan, PhD
Associate Editor
Communications Biology
orcid.org/0000-0002-7369-9225

Reviewers' comments:

Reviewer #1 (Remarks to the Author):

The study looked at changes in brain connectivity estimated using fMRI and various blood markers pre and post a 7-day retreat which comprised of meditation, reconceptualization and healing. The study is interesting as it looks at an integrative way on how different brain body markers are changing.

Apart from the small sample size which is understandable given the large amount of data collected per individual, the study is well done. I commend the authors for their integrative approach in investigating the mechanisms underlying meditation retreats examining both brain changes and various blood markers.

I have some minor comments as follows:

1. 5 minutes of resting state data is usually not sufficient to make strong claims about brain networks and properties given the variability in FC measures based on various brain states and other physiological properties (Laumann et al., 2024). Please mention these limitations in the discussion section
2. Given that the proteomic and metabolomic analysis were hypothesis free and data driven, shouldn't corrections for multiple comparisons be applied to limit the False Discovery Rate?
3. Figure 9: It's not only predictive coding which seems to have an impact on brain body axis but different meditation techniques are impacting the sympathetic and parasympathetic nervous system (Sezer and Sacchet, 2025) which is having an impact on various blood biomarkers.
4. Methods:
 - a. More details should be added here about the participants – what kind of meditation practice they have done, total years of practice and meditation experience. Also, please add more details about the meditation practices done in the retreat. Were they guided or not?

b. Was there only one rest and one meditation run per session?

References:

Laumann, T. O., Snyder, A. Z., Gratton, C. (2024). Challenges in the measurement and interpretation of dynamic functional connectivity. *Imaging Neuroscience*.
Sezer, I., & Sacchet, M. D. (2025). Advanced and long-term meditation and the autonomic nervous system: A review and synthesis. *Neuroscience and Biobehavioral Reviews*, 173, 1–25. <https://doi.org/10.1016/j.neubiorev.2025.106141>

Reviewer #2 (Remarks to the Author):

This manuscript presents an original and methodologically rigorous investigation into the combined neural and molecular effects of a mind-body retreat incorporating meditation, reconceptualization, and open-label placebo-like rituals. While the sample size is relatively small, the study is well-executed and transparent about its limitations. I thank the authors for this thoughtful and ambitious manuscript.

Introduction

“Meditation... can produce subjective mystical-type experiences (10) associated with mental health improvements (11, 12)...”. While Zanesco et al. appropriately support the claim in a meditative context, the inclusion of psychedelic literature in this statement (Refs 11 and 12) introduces confusion. Psychedelic-induced mystical states may share surface-level phenomenology with meditation-induced experiences, but they differ significantly in terms of neurochemical pathways. The authors should clearly distinguish between mystical experiences induced by meditation versus psychedelics.

Methods

Could the authors justify the eyes-open instruction for meditation, given its potential to introduce visual confounds in BOLD signal interpretation?

The authors applied framewise displacement (FD) filtering. Please provide, per participant, the number or proportion of frames excluded for the resting state and meditation scans (e.g., in a supplementary table). Also, confirm whether the FD values reported in Supplementary Table S1 reflect values before or after frame exclusion.

The use of ICA-AROMA and regression of CSF/WM signals addresses motion and some physiological noise. However, RETROICOR or aCompCor were not mentioned. Please confirm whether physiological recordings (e.g., cardiac/respiratory belts) were collected, or explicitly state their absence.

Were all network ROIs derived from a single consistent parcellation (Raichle atlas), except for the eight task-based ROIs from DiFuMo? If so, what was the rationale for introducing additional ROIs from DiFuMo, especially when regions like mPFC and other DMN nodes already exist in the Raichle atlas?

“...Pearson correlations were calculated as measures of connectivity strength.” Were these correlations Fisher z-transformed before statistical analysis? If not, this may violate parametric assumptions.

The authors may want to comment on potential confounds introduced by variable pre-collection fasting durations and collection times for plasma e.g., circadian or metabolic influences (“Day -1”, 2 days pre-retreat, 2-4 pm) or on-site (“Day 0”, 1 day pre-retreat, 11 am-4 pm) and post-intervention (“Day 8”, on-site, 9-5 pm).

Many of the plasma collection and proteomics procedures are described in sufficient technical detail, but they would benefit from references to established protocols or prior studies to support methodological choices. This applies especially to subsections such as Cell Culture and Plasma Treatment, Seahorse XF Assays, Live-Cell Imaging and Neurite Analysis, and Exosome-Specific Small RNA Transcriptomics.

Results / Discussions / Limitations

Please clarify whether any psychedelic substances were involved in the retreat, and explicitly state their absence if they were not included.

The finding of reduced connectivity between medial prefrontal cortex (mPFC) and cerebellar ROIs during meditation (observed both pre- and post-intervention) is particularly interesting. This likely reflects a state-dependent suppression of self-referential and evaluative processing. Could this effect be linked specifically to the features of the guided meditation protocol, such as interoceptive focus (e.g., breathing or body scan)? It would be helpful if the authors could clarify the instructions and reference any prior studies that report similar cerebello–prefrontal decoupling during meditation.

Given the sensitivity of plasma proteomic and metabolomic measures to recent dietary intake, please clarify whether any specific dietary restrictions or fasting protocols were followed during the retreat. If no standardized diet was implemented, this should be explicitly acknowledged as a potential confound. This information is important to contextualize the observed molecular changes.

The authors report simultaneous upregulation of both inflammatory and anti-inflammatory proteins post-intervention. This is intriguing, the manuscript would benefit from an interpretation of how these seemingly opposing shifts should be understood in context of prior literature.

Were any of the DMN-FC, plasma or omics changes correlated with self-report measures (e.g., MEQ, mood, stress)? Even exploratory analyses here could help link molecular changes to psychological outcomes and ground the biological results in participants’ subjective experience.

** See the Nature Portfolio author and referees' website at www.nature.com/authors for information about policies, services and author benefits

Communications Biology is committed to improving transparency in authorship. As part of our efforts in this direction, we are now requesting that all authors identified as 'corresponding author' create and link their Open Researcher and Contributor Identifier (ORCID) with their account on the Manuscript Tracking System prior to acceptance. ORCID helps the scientific community achieve unambiguous attribution of all scholarly contributions. You can create and link your ORCID from the home page of the Manuscript Tracking System by clicking on 'Modify my Springer Nature account' and following the instructions in the link below. Please also inform all co-authors that they can add their ORCIDs to their accounts and that they must do so prior to acceptance.

Version 1:

Decision Letter:

** Please ensure you delete the link to your author homepage in this email if you wish to forward it to your coauthors **

Dear Dr Patel,

Your manuscript entitled "Neural and Molecular Changes During a Mind-Body Reconceptualization, Meditation, and Open Label Placebo Healing Intervention" has now been seen again by our referees, whose comments appear below. In light of their advice I am delighted to say that we are happy, in principle, to publish a suitably revised version in Communications Biology.

We therefore invite you to revise your paper one last time to comply with our format requirements and to maximise the accessibility and therefore the impact of your work.

* Please see the attached document for editorial requests for the final version (.docx file). Please ensure a completed version of this file is uploaded as a Related Manuscript with your final submission.

* Please review our [final submission file checklist](https://www.nature.com/documents/commsj-file-checklist.pdf) to ensure all necessary files are present with your final submission and to avoid delays in accepting your manuscript. For your reference, a style and formatting guide is available [here](https://www.nature.com/documents/commsj-life-style-formatting-guide-accept.pdf) and includes all of our style requirements.

It is important that you pay careful attention to the requests in these documents to avoid a delay in formal acceptance of the article.

Open access

Communications Biology is a fully open access journal. Articles are made freely accessible on publication. For further information about article processing charges, open access funding, and advice and support from Nature Research, please visit <https://www.nature.com/commsbio/open-access>

Please use the following link to upload your revised files:

Link Redacted

We hope to hear from you within two weeks. If you expect the process to take longer than one month, please let us know.

Congratulations on an excellent paper!

Best regards,

Jasmine Pan, PhD
Associate Editor
Communications Biology
orcid.org/0000-0002-7369-9225

PS: At acceptance, the corresponding author will be provided with instructions for completing the license on behalf of all authors. This grants us the necessary permissions to publish your paper. Additionally, you will be asked to declare that all required third party permissions have been obtained, and to provide billing information in order to pay the article-processing charge (APC).

REVIEWERS' COMMENTS:

Reviewer #1 (Remarks to the Author):

The authors have addressed all my comments. Good job with the manuscript.

Reviewer #2 (Remarks to the Author):

This manuscript presents an original and methodologically rigorous investigation into the combined neural and molecular effects of a mind-body retreat incorporating meditation and placebo-like rituals. While the sample size is relatively small, the study is well-executed and transparent about its limitations. The revised manuscript has addressed all of my prior concerns comprehensively. The authors have clarified their methodological choices, added several appropriate references, expanded the discussion of limitations, and conducted additional analyses, including an analysis comparing subjective outcomes to biological measures. The manuscript contains sufficient methodological detail to enable reproducibility, particularly after the authors' clarifications.

Overall, the revisions have substantially strengthened the manuscript, and I have no further concerns. I believe it is now suitable for publication.

** See the Nature Portfolio author and referees' website at www.nature.com/authors for information about policies, services and author benefits

Reviewer #1 (Remarks to the Author):

The study looked at changes in brain connectivity estimated using fMRI and various blood markers pre and post a 7-day retreat which comprised of meditation, reconceptualization and healing. The study is interesting as it looks at an integrative way on how different brain body markers are changing.

Apart from the small sample size which is understandable given the large amount of data collected per individual, the study is well done. I commend the authors for their integrative approach in investigating the mechanisms underlying meditation retreats examining both brain changes and various blood markers.

I have some minor comments as follows:

1. 5 minutes of resting state data is usually not sufficient to make strong claims about brain networks and properties given the variability in FC measures based on various brain states and other physiological properties (Laumann et al., 2024). Please mention these limitations in the discussion section

The authors thank the reviewer for this pertinent comment. The discussion section has been amended to include the following statement (lines 633-634): “The short duration of the resting state scan can limit our ability to attribute connectivity-derived features solely to the experimental variables (Laumann et al., 2024).”

2. Given that the proteomic and metabolomic analysis were hypothesis free and data driven, shouldn't corrections for multiple comparisons be applied to limit the False Discovery Rate?

Thank you for this important point. We agree that in hypothesis-free, data-driven analyses, controlling for multiple comparisons is essential to limit false positives. We applied False Discovery Rate (FDR) correction using the Benjamini–Hochberg procedure to our primary analyses. Results are reported as FDR-adjusted significance thresholds, with nominal p-values included for transparency. We also added the following statement to the Methods section under “Statistics” (lines 1074-1076): “Given that proteomic and metabolomic analyses were hypothesis free and data driven, False Discovery Rate (FDR) correction was applied using the Benjamini-Hochberg procedure, and reported p-values are FDR-adjusted, with nominal p-values included for transparency.” We added further detail in the Metabolomics Methods section as follows (lines 918-922): “Pathway enrichment was first assessed using MetaboAnalyst, and false discovery rate (FDR) correction was applied to control for multiple comparisons. Significant pathways ($p_{\text{FDR}} < 0.05$) were then selected for follow-up analysis. For these pathways, we examined the distribution of normalized feature values and report the corresponding p-values for group comparisons after normalization.” The legend of Figure 6 was amended to clearly differentiate between FDR-adjusted

p-values for metabolomic pathways and nominal p-values for metabolite changes within the FDR-significant tryptophan pathway (line 365).

3. Figure 9: It's not only predictive coding which seems to have an impact on brain body axis but different meditation techniques are impacting the sympathetic and parasympathetic nervous system (Sezer and Sacchet, 2025) which is having an impact on various blood biomarkers.

This is an important point of discussion, and the authors thank the reviewer for pointing it out. The figure has been edited to include the autonomic pathway and discussion text has been amended to include the following (lines 558-563): "The observed changes in blood biomarkers may have resulted from the endocrine, immune, and other regulatory changes evoked by the intervention's effects on these predictive allostatic mechanisms, as well as from meditation-induced effects on both sympathetic and parasympathetic branches of the autonomic nervous system, a finding common to other meditation studies (Sezer and Sacchet, 2025), which may also account for the concomitant upregulation of inflammatory and anti-inflammatory proteins during the intervention." The figure legend (lines 541-542) has been edited to include discussion of sympathetic and parasympathetic activation as a result of meditation. Whether meditation-driven autonomic effects are mediated by allostatic predictive mechanisms, changes in breathing patterns, or other means is a topic of research interest for the authors.

4. Methods:

a. More details should be added here about the participants – what kind of meditation practice they have done, total years of practice and meditation experience.

The authors acknowledge the importance of these details and thank the reviewer for these helpful comments. While each study participant was asked specifically about the length of practice and the frequency of the retreat meditations in the pre-study survey, they were only asked whether or not they had another meditation practice. Thus, length and frequency were not established for “other” meditation practices. We do realize that the lack of this information and the heterogeneity of meditation practices and experience levels are a confound and have added a sentence to our discussion to highlight this important limitation (lines 624-627): “Although data on meditation experience and practice frequency were collected for retreat-based meditations within the study cohort, equivalent information regarding other forms of meditation practice—apart from the binary indication of their presence or absence—was not obtained.”

We have also expanded the Methods section under “Participants” to include details of meditation experience and frequency of practice of the retreat meditations, as well as details on how many of the participants were practitioners of other forms of meditation (lines 665-674):

“Of the 6 male participants, 2 had been practicing the meditations carried out in the workshop daily for more than 1 year, 3 had been practicing the meditations daily for less than 1 year (one of whom practiced a different form of meditation), and 1 participant had not previously practiced the meditations before attending the retreat, though he had a different meditation practice. Of the 14 female participants, 8 had been practicing the retreat meditations for over 1 year (most being a daily practice), and 1 of the 8 also practiced a different form of meditation. Six of the 8 females had been practicing the meditations for less than 1 year, with only 3 reporting an almost daily practice, and 4 of the 8 reported having a different meditation practice. One of the 8 females had not previously practiced the meditations before attending the retreat, though she had a different meditation practice.”.

Also, please add more details about the meditation practices done in the retreat. Were they guided or not?

The authors acknowledge the importance of a more detailed description of the meditation practice, especially in light of the great diversity of meditation techniques and the heterogeneity in the scientific literature. Further details about the guided element and the techniques themselves have been added to the intervention description in the introduction as follows (lines 71-78):

“All meditations (33 total hours) were guided, delivered with atmospheric music, and taught Kundalini techniques, which combine conscious meta-awareness and conscious breathing exercises with slow, ascending, focused interoceptive attention on purported energetic centers along the midline (e.g., brow, throat, heart) which, according to practitioners, can reprocess embodied trauma and catalyze adaptive mental and physical changes (27, 28). Guidance also emphasized sustaining a heart-centered state

devoid of thinking or judgment and focusing awareness on a void beyond one's normal sense of space and time – a common theme in some contemplative practices (29)”

In the methods section, description of the fMRI meditation audio was amended to include further details of the meditation as follows (lines 715-718): “The audio consisted of continuous expansive atmospheric music with an intermittent ethereal voice repeatedly instructing listeners to “tune into nothingness”, “no time”, “nowhere”, and “love”, combining loving-kindness meditation with focused awareness of the experiencing self and its potential dissolution into pure awareness.”

b. Was there only one rest and one meditation run per session?

Yes, and this has been made explicit during the fMRI functional scans methods section, where the following statement was added (lines 714-715): “One rest and one meditation run were collected per scan.”

Reviewer #2 (Remarks to the Author):

This manuscript presents an original and methodologically rigorous investigation into the combined neural and molecular effects of a mind-body retreat incorporating meditation, reconceptualization, and open-label placebo-like rituals. While the sample size is relatively small, the study is well-executed and transparent about its limitations. I thank the authors for this thoughtful and ambitious manuscript.

5. Introduction

“Meditation... can produce subjective mystical-type experiences (10) associated with mental health improvements (11, 12)...”. While Zanesco et al. appropriately support the claim in a meditative context, the inclusion of psychedelic literature in this statement (Refs 11 and 12) introduces confusion. Psychedelic-induced mystical states may share surface-level phenomenology with meditation-induced experiences, but they differ significantly in terms of neurochemical pathways. The authors should clearly distinguish between mystical experiences induced by meditation versus psychedelics.

The authors thank the reviewer for this pertinent comment, as the inclusion of references to psychedelics introduced unnecessary confusion and fell outside the scope of this paper. Citations and references to psychedelics and psychedelic-induced states have been removed and replaced with meditation-specific references in both the introduction (lines 41-42 and 88) and discussion sections (lines 567, 570), including von Lutterveld et al. (2017) and Brewer et al. (2011), referencing changes in DMN and whole-brain network functional connectivity.

References:

van Lutterveld R, van Dellen E, Pal P, Yang H, Stam CJ, Brewer J. Meditation is associated with increased brain network integration. *Neuroimage*. 2017 Sep;158:18-25. doi: 10.1016/j.neuroimage.2017.06.071. Epub 2017 Jun 27. PMID: 28663069.

Brewer JA, Worhunsky PD, Gray JR, Tang YY, Weber J, Kober H. Meditation experience is associated with differences in default mode network activity and connectivity. *Proc Natl Acad Sci U S A*. 2011 Dec 13;108(50):20254-9. doi: 10.1073/pnas.1112029108. Epub 2011 Nov 23. PMID: 22114193.

6. Methods

Could the authors justify the eyes-open instruction for meditation, given its potential to introduce visual confounds in BOLD signal interpretation?

The authors thank the reviewer for bringing up this important issue, a matter of internal debate during experimental design. Eyes-open was ultimately chosen to avoid the risk of participants entering a non-wakeful state and producing sleep-related artifacts, especially in light of the 15-minute-long meditation scan and the rigorous and potentially tiresome retreat schedule, which involved 4am wake-up days. The following justification was added to the methods section (lines 711-714): “An eyes open paradigm was chosen to reduce the risk of drowsiness or sleep during the meditation run, which was deemed a potentially greater risk than the presence of visual confounds to accurately compare active meditation with passive rest (Garrison et al. 2013; Tagliazucchi & Laufs 2014).”

The potential for visual confounds was also added as a potential limitation in the discussion section as follows (lines 634-636): “The open eyes paradigm during fMRI BOLD acquisition, chosen to reduce the risk of drowsiness, likely introduced potential visual confounds.”

References:

Garrison KA, Santoyo JF, Davis JH, Thornhill TA 4th, Kerr CE, Brewer JA. Effortless awareness: using real time neurofeedback to investigate correlates of posterior cingulate cortex activity in meditators' self-report. *Front Hum Neurosci*. 2013 Aug 6;7:440. doi: 10.3389/fnhum.2013.00440. PMID: 23964222.

Tagliazucchi E, Laufs H. Decoding wakefulness levels from typical fMRI resting-state data reveals reliable drifts between wakefulness and sleep. *Neuron*. 2014 May 7;82(3):695-708. doi: 10.1016/j.neuron.2014.03.020. PMID: 24811386.

7. The authors applied framewise displacement (FD) filtering. Please provide, per participant, the number or proportion of frames excluded for the resting state and meditation scans (e.g., in a supplementary table). Also, confirm whether the FD values reported in Supplementary Table S1 reflect values before or after frame exclusion.

Motion artifacts were removed with ICA-AROMA regression only. The denoising methods did say that “confounding time-series were calculated for framewise displacement (FD), DVARS, and CSF/WM region-wise global signals”; however, the FD confound time series were not *regressed* since the minimally-distortive denoising pipeline (ICA-AROMA + CSF/WM) removed motion adequately. To avoid confusion, lines 738-740 were edited to: “Confounding time-series were calculated for CSF/WM region-wise global signals, and motion artifacts were identified using independent component analysis (ICA-AROMA).” To assess whether motion artifacts were successfully removed for functional connectivity analysis, functional connectivity value distributions per BOLD run were plotted and visually inspected to be mean-centered around 0. Only one outlier run was found to be significantly de-centered, and the whole run was excluded from further analysis. Our main concern was that average motion is higher for meditation than rest runs (a limitation noted in the manuscript), so FD scrubbing would lead to more frames removed from meditation than rest runs, potentially reducing motion artifacts beyond ICA-AROMA but also likely biasing meditation versus rest comparisons. Given the satisfactory QC visual assessment of the chosen denoising pipeline, we decided not to perform FD scrubbing/filtering or any further confound regression to avoid bias and maximize degrees of freedom available. Since mentioning that FD confound time series were calculated and then not saying anything further about them confuses the reader into believing that FD filtering was performed, we removed mention of FD confound time series calculation from the methods.

8. The use of ICA-AROMA and regression of CSF/WM signals addresses motion and some physiological noise. However, RETROICOR or aCompCor were not mentioned. Please confirm whether physiological recordings (e.g., cardiac/respiratory belts) were collected, or explicitly state their absence.

We thank the reviewer for pointing our attention to this omission. We have added explicit mention of not using physiological recordings in the fMRI methods (lines 744-745): “Physiological noise was addressed via ICA-AROMA + CSF/WM regressions rather than with physiological recordings”. We also added a discussion of the potential of ICA-AROMA + CSF/WM regression to not adequately remove physiological noise in regions with strong cardiac or respiratory activity as a limitation of the paper in the discussion section (lines 636-638): “[D]enoising with ICA-AROMA + WM/CSF regression is limited in removing physiological noise in regions with strong cardiac or respiratory activity, which may have resulted in additional confounds.”

9. Were all network ROIs derived from a single consistent parcellation (Raichle atlas), except for the eight task-based ROIs from DiFuMo? If so, what was the rationale for introducing additional ROIs from DiFuMo, especially when regions like mPFC and other DMN nodes already exist in the Raichle atlas?

As noted, the 36 ROIs for the exploratory, hypothesis-free resting state network analyses were derived from a single consistent parcellation (Raichle atlas). Since statistical tests for ROI-ROI connectivity for these would be corrected for 630 multiple comparisons, we opted a priori to include a separate, hypothesis-driven analysis with a narrower set of 8 ROIs that were affected by meditation in previous studies. Since some of these areas were not represented in the Raichle atlas, a separate atlas that contained all 8 (DiFuMo) was chosen. We were glad to find that several significant meditation-driven ROI-ROI FC changes survived correction for 630 multiple comparisons in the hypothesis-free analysis (Figure 2B). This led us to privilege these results over the more limited 8 ROIs, which agree with the hypothesis-free ones but are not given as much attention in the manuscript. In the methods section (“Resting State Networks and Regions of Interest”), we clarified that the rationale behind additionally including these 8 was for a hypothesis-driven analysis (line 770). We also emphasized in the fMRI Results section that the a priori ROIs served as an internal replication, given that they returned results agreeable with those from the RSN data-driven approach (lines 152-153): “ROIs also revealed lower functional connectivity between key network hubs, including medial prefrontal cortex (mPFC), precuneus, bilateral insular cortices, and bilateral angular gyri (Supplementary Table S5), serving as an internal replication and returning results agreeable with the RSN data-driven approach.”.

10. “...Pearson correlations were calculated as measures of connectivity strength.” Were these correlations Fisher z-transformed before statistical analysis? If not, this may violate parametric assumptions.

The authors thank the reviewer for pointing out this important methodological point. Correlation values had not been Fisher z-transformed, which was statistically problematic. The functional connectivity analysis was repeated with Fisher z-transformed values, applied with numpy’s `np.arctanh(r)` function. All relevant results, figures, table values (lines 131-176), and methods (lines 766, 783) have been updated. No major changes in the RSN FC contrasts or results were noticed, and all values now reflect updated and statistically correct Cohen’s *d*, *t*, and *p* values.

11. The authors may want to comment on potential confounds introduced by variable pre-collection fasting durations and collection times for plasma e.g., circadian or metabolic influences (“Day -1”, 2 days pre-retreat, 2-4 pm) or on-site (“Day 0”, 1 day pre-retreat, 11 am-4 pm) and post-intervention (“Day 8”, on-site, 9-5 pm).

The authors thank the reviewer for pointing out this important potential set of confounds. The following was added to the discussion of study limitations to comment upon them (lines 627-630): “Potential circadian and metabolic confounds were introduced by variable blood collection times (up to 8 hour range on Day 8), pre-collection fasting durations, and time elapsed after the meditation intervention ended (up to 48 hours for blood collections and 24 hours for fMRI acquisition) between participants.”

12. Many of the plasma collection and proteomics procedures are described in sufficient technical detail, but they would benefit from references to established protocols or prior studies to support methodological choices. This applies especially to subsections such as Cell Culture and Plasma Treatment, Seahorse XF Assays, Live-Cell Imaging and Neurite Analysis, and Exosome-Specific Small RNA Transcriptomics.

We appreciate the reviewer’s suggestion and have added references to well-established protocols and resources to substantiate methodological choices:

Cell Culture and Plasma Treatment:

Zuniga-Hertz JP, *et al.* Multidimensional Analysis of Twin Sets During an Intensive Week-Long Meditation Retreat: A Pilot Study. *Mindfulness* (N Y). 2025;16(6):1634-1655. doi: 10.1007/s12671-025-02584-x. Epub 2025 May 7. PMID: 40535580; PMCID: PMC12170707.

Song Y, *et al.* Impact of blood factors on endothelial cell metabolism and function in two diverse heart failure models. *PLoS One*. 2023 Feb 13;18(2):e0281550. doi: 10.1371/journal.pone.0281550. PMID: 36780477; PMCID: PMC9924994.

Seahorse XF Assays:

Kopp EL, *et al.* Modeling and Phenotyping Acute and Chronic Type 2 Diabetes Mellitus In Vitro in Rodent Heart and Skeletal Muscle Cells. *Cells*. 2023 Dec 7;12(24):2786. doi: 10.3390/cells12242786. PMID: 38132105; PMCID: PMC10741513.

Live-Cell Imaging and Neurite Analysis:

Karliner J, Merry DE. Differentiating PC12 cells to evaluate neurite densities through live-cell imaging. *STAR Protocol*. 2023 Mar 17;4(1):101993. doi: 10.1016/j.xpro.2022.101993. Epub 2023 Jan 4. PMID: 36602900; PMCID: PMC9826846.

Exosome-Specific Small RNA Transcriptomics:

Martin M. Cutadapt removes adapter sequences from high-throughput sequencing reads. *EMBnet.journal*. Vol 17, No 1. (Year not specified); DOI: 10.14806/ej.17.1.200.

Dobin A, Davis CA, Schlesinger F, *et al.* STAR: ultrafast universal RNA-seq aligner. *Bioinformatics*. 2013 Jan 1;29(1):15-21. doi: 10.1093/bioinformatics/bts635. Epub 2012 Oct 25. PMID: 23104886.

Ewels P, Magnusson M, Lundin S, Källér M. MultiQC: summarize analysis results for multiple tools and samples in a single report. *Bioinformatics*. 2016 Oct 1;32(19):3047-3048. doi: 10.1093/bioinformatics/btw354. Epub 2016 Jun 16. PMID: 27312411.

McCarthy DJ, Chen Y, Smyth GK. Differential expression analysis of multifactor RNA-Seq experiments with respect to biological variation. *Nucleic Acids Res*. 2012 May;40(10):4288-4297. doi: 10.1093/nar/gks042. Epub 2012 Jan 28. PMID: 22287627.

Kozomara A, Birgaoanu M, Griffiths-Jones S. miRBase: from microRNA sequences to function. *Nucleic Acids Res*. 2019 Jan 8;47(D1):D155-D162. doi: 10.1093/nar/gky1141. PMID: 30423142.

Robinson MD, Oshlack A. A scaling normalization method for differential expression analysis of RNA-seq data. *Genome Biol*. 2010;11(3):R25. doi: 10.1186/gb-2010-11-3-r25. Epub 2010 Mar 2. PMID: 20196867.

Results / Discussions / Limitations

13. Please clarify whether any psychedelic substances were involved in the retreat, and explicitly state their absence if they were not included.

No psychedelic or other pharmacological substances were involved in the retreat. We have added an explicit statement of this in the introduction alongside the description of the intervention (line 85): “No pharmacological substances, including any psychedelics, were involved in the retreat.”

14. The finding of reduced connectivity between medial prefrontal cortex (mPFC) and cerebellar ROIs during meditation (observed both pre- and post-intervention) is particularly interesting. This likely reflects a state-dependent suppression of self-referential and evaluative processing. Could this effect be linked specifically to the features of the guided meditation protocol, such as interoceptive focus (e.g., breathing or body scan)? It would be helpful if the authors could clarify the instructions and reference any prior studies that report similar cerebello–prefrontal decoupling during meditation.

This finding is indeed interesting and merits further discussion in the paper. We thank the reviewer for pointing it out. The guided meditation protocol featured in the introduction have been expanded to include

(lines 71-75) “conscious meta-awareness [and] conscious breathing exercises with slow, ascending, focused interoceptive attention on purported energetic centers along the midline (e.g., brow, throat, heart)” to make it clear that there was a significant component of interoceptive focus, as well as (lines 76-78): “guidance also emphasized sustaining a heart-centered state devoid of thinking or judgment and focusing awareness on a void beyond one’s normal sense of space and time – a common theme in some contemplative practices,” to emphasize that the meditative state likely did involve suppression of self-referential and evaluative processing.

An interpretation of DMN and prefrontal-cerebellar FC changes in light of the guided meditation features and with reference to previous studies was added to the discussion section as follows (lines 512-522): “[Disruption of DMN and cerebellar-prefrontal connectivity] complements studies showing meditation-induced changes in DMN connectivity (Guidotti et al. 2023). Mindfulness meditation has been linked to DMN deactivation (Brewer et al., 2011; Garrison et al., 2015) and stronger connectivity between DMN-Salience and DMN-Executive Control networks (Bremer et al. 2022), likely reflecting an increased capacity to switch in/out of default mode dominance. Here, the main effect we observed was a meditation-driven decrease in intra-network DMN connectivity and a broad desynchronization of whole-brain connectivity. Meditation also reduced prefrontal-cerebellar connectivity both pre- and post-intervention, suggesting a state-dependent suppression of self-referential and evaluative processing consistent with focused interoceptive awareness, non-judgmental awareness, and a state that transcends the self – all features of the guided meditations. DMN-cerebellar connectivity alterations have been reported for depressive disorder and during previous meditation studies, albeit with opposite (higher connectivity) effects (Shen et al. 2020; Guo et al. 2013).”

References

Bremer B, *et al.* Mindfulness meditation increases default mode, salience, and central executive network connectivity. *Sci Rep.* 2022 Aug 2;12(1):13219. doi: 10.1038/s41598-022-17325-6. PMID: 35918449; PMCID: PMC9346127.

Brewer JA, *et al.* Meditation experience is associated with differences in default mode network activity and connectivity. *Proc Natl Acad Sci U S A.* 2011 Dec 13;108(50):20254-9. doi: 10.1073/pnas.1112029108. Epub 2011 Nov 23. PMID: 22114193; PMCID: PMC3250176.

Garrison KA, *et al.* Meditation leads to reduced default mode network activity beyond an active task. *Cogn Affect Behav Neurosci.* 2015 Sep;15(3):712-20. doi: 10.3758/s13415-015-0358-3. PMID: 25904238; PMCID: PMC4529365.

Guidotti R, *et al.* Long-Term and Meditation-Specific Modulations of Brain Connectivity Revealed Through Multivariate Pattern Analysis. *Brain Topogr.* 2023 May;36(3):409-418. doi: 10.1007/s10548-023-00950-3. Epub 2023 Mar 28. PMID: 36977909; PMCID: PMC10164028.

Guo W, *et al.* Abnormal resting-state cerebellar-cerebral functional connectivity in treatment-resistant depression and treatment sensitive depression. *Prog Neuropsychopharmacol Biol Psychiatry.* 2013 Jul 1;44:51-7. doi: 10.1016/j.pnpbp.2013.01.010. Epub 2013 Jan 23. PMID: 23352887.

Shen, Y.-Q., *et al.* (2020). "Meditation effect in changing functional integrations across large-scale brain networks: Preliminary evidence from a meta-analysis of seed-based functional connectivity": Corrigendum. *Journal of Pacific Rim Psychology*, 14, Article e22. <https://doi.org/10.1017/prp.2020.16>

15. Given the sensitivity of plasma proteomic and metabolomic measures to recent dietary intake, please clarify whether any specific dietary restrictions or fasting protocols were followed during the retreat. If no standardized diet was implemented, this should be explicitly acknowledged as a potential confound. This information is important to contextualize the observed molecular changes.

We thank the reviewer for highlighting dietary intake as a potential confounder. We have acknowledged this in the Discussion section and added the following information to the Methods ("Human Plasma Collection") section (lines 821-823): "Participants were offered the same food options for all breakfast, lunch, and snacks throughout the week, although food choices were not monitored. All participants were required to fast for at least 30 minutes before blood collection."

While no standardized diet or fasting protocol was implemented during the retreat, we sought to minimize diet-related artifacts during data processing. Specifically, in the metabolomics pipeline, we leveraged HMDB annotations available through MetaboAnalyst, which flag metabolites classified as dietary or drug-derived exogenous compounds. These annotations were systematically checked, and any such metabolites that appeared among the top 25 features were flagged and, if necessary, removed from downstream analyses. This process is now described in the Methods ("Metabolite Analysis") section (lines 909-913): "Dietary factors and fasting may have introduced confounds in proteomic and metabolomic measures, as no standardized diet was implemented, and fasting times beyond 30 minutes pre-blood collection were not controlled. Exogenous compounds annotated as dietary or drug-related by HMDB were monitored using the MetaboAnalyst pipeline and were flagged or removed if observed among the top 25 features."

16. The authors report simultaneous upregulation of both inflammatory and anti-inflammatory proteins post-intervention. This is intriguing, the manuscript would benefit from an interpretation of how these seemingly opposing shifts should be understood in context of prior literature.

Indeed, this finding was interesting, and we thank the reviewer for highlighting it. Our group recently published a pilot study in a small meditating twin cohort (Zuniga-Hertz JP, 2025, PMID 40535580, ref. above). Conducted in parallel with the present study, that work used an exploratory panel of cytokines, biologically active enzymes, and growth factors. As in the present study, we observed a simultaneous increasing trend in both pro- and anti-inflammatory proteins, with an overall pattern suggesting enhanced tissue repair and regeneration. Complementary metabolomic studies in the same cohort further support this conclusion.

This dual regulation is consistent with prior literature showing that inflammatory mediators can act as alarmins to initiate immune activation, while also promoting compensatory regulatory pathways. For example, S100A8 and S100A9 act as endogenous damage-associated molecular patterns that induce secretion of cytokines such as IL-6, IL-8, and CCL2 (Crowe et al., 2019; Cremers et al., 2017). At the same time, anti-inflammatory regulators such as SOCS3 and TNFAIP3 are often co-induced to dampen NF- κ B signaling and restore homeostasis (Vogl et al., 2018). Similarly, cytokines like IL-6 are known to have both pro- and anti-inflammatory roles depending on context—acting as an acute-phase reactant during stress, but also stimulating IL-1 receptor antagonist and IL-10 in exercise and repair settings (Vadaq et al., 2022).

Taken together, the concurrent upregulation of inflammatory and anti-inflammatory proteins in our cohort is best understood as evidence of a coordinated, adaptive immune response. The intervention appears to initiate controlled immune activation, while simultaneously engaging counter-regulatory pathways to limit excessive inflammation and support tissue turnover and repair. In the manuscript, the following text was added (lines 605-609: “Interestingly, simultaneous activation of inflammatory and anti-inflammatory protein pathways is reminiscent of prior findings showing mind-body techniques can enhance resilience to environmental stressors via gene expression (64). Cytokines like IL-6 are known to have both pro-and anti-inflammatory roles depending on the context acting as both acute-phase reactants during stress but also stimulating IL-1 receptor antagonist and IL-10 in exercise and repair settings (Vadaq et al., 2022).

References

Cremers, N.A.J. *et al.* S100A8/A9 increases the mobilization of pro-inflammatory Ly6Chigh monocytes to the synovium during experimental osteoarthritis. *Arthritis Res Ther* 19, 217 (2017).
<https://doi.org/10.1186/s13075-017-1426-6>

Crowe LAN, *et al.* S100A8 & S100A9: Alarmin mediated inflammation in tendinopathy. *Sci Rep.* 2019 Feb 6;9(1):1463. doi: 10.1038/s41598-018-37684-3. PMID: 30728384; PMCID: PMC6365574.

Vadaq N, *et al.* Targeted plasma proteomics reveals upregulation of distinct inflammatory pathways in people living with HIV. *iScience.* 2022 Sep 7;25(10):105089. doi: 10.1016/j.isci.2022.105089. PMID: 36157576; PMCID: PMC9494231.

Vogl T, *et al.* Autoinhibitory regulation of S100A8/S100A9 alarmin activity locally restricts sterile inflammation. *J Clin Invest.* 2018 May 1;128(5):1852-1866. doi: 10.1172/JCI89867. Epub 2018 Apr 3. PMID: 29611822; PMCID: PMC5919817.

17. Were any of the DMN-FC, plasma or omics changes correlated with self-report measures (e.g., MEQ, mood, stress)? Even exploratory analyses here could help link molecular changes to psychological outcomes and ground the biological results in participants' subjective experience.

We appreciate the reviewer's insightful suggestion. To address this, we performed additional exploratory correlations between the machine learning-identified features and self-reported outcomes (MEQ). These analyses are now included in Figure 8C. fMRI features were also added to the ML modeling.

As shown, several metabolites, proteins, and functional connectivity features demonstrated significant associations with MEQ scores (FDR-corrected $p < 0.05$, denoted by asterisks). For example, at baseline, 3-indolebutyric acid was strongly positively correlated with MEQ. Post-intervention, N-acetyl-D-phenylalanine and glycyl-lysine were positively associated with MEQ. In the experience-based analyses, FGF19 and interferon gamma exhibited significant negative correlations with MEQ delta scores, while proteins such as neurotrophin 4 showed positive associations post-intervention.

These exploratory results help to ground the molecular and brain changes in participants' subjective experiences, highlighting potential links between biological mechanisms and self-reported outcomes. We have added this description to the Results section and included the new Figure 8C, with additions to the Discussion section to consider the caveats with comparing subjective outcomes to biological measures.

The following text was added at lines 468-502:

“Focusing first on timepoint classification, both models achieved strong discrimination between pre- and post-meditation states (XGBoost AUC = 0.86; Random Forest AUC = 0.90). Top XGBoost predictors included Gly-Gly-Phe, 3-indolebutyric acid, Gly-Lys, and connectivity between the anterior and salience networks, default mode–salience, and precuneus regions. Random Forest highlighted overlapping features alongside additional metabolites such as Androstenediol, S1P(d18:1), N6-methyladenosine, and purine derivatives. Many of these metabolites are linked to amino acid metabolism, neurotransmitter balance, and cellular stress signaling, while sphingolipids such as Androstenediol and S1P(d18:1) implicate lipid signaling and neuroendocrine function. Connectivity difference involving salience, executive and default mode networks (SN, ECN, DMN,PC) align with prior literature that mediation can reorganize large-scale brain networks that govern salience (Bremer et al. 2022), attention (Vishnubhotla et al., 2021), interoception (Datko et al., 2022), and self-referential processing (Bremer et al. 2022). Taken in concert, timepoint differences identify shifts that integrate across neurofunctional and metabolic layers, reinforcing the concept that mediation acts as a systemic regulator of mind-body physiology.

To evaluate the influence of meditation experience, post–pre deltas were compared between novice and advanced practitioners. XGBoost achieved modest classification performance (AUC = 0.70), while Random Forest reached higher discrimination (AUC = 0.93). SHAP analysis indicated that immune- and stress-related proteins, including fibroblast growth factor 19, interferon- γ , C-C motif chemokine ligand 5, and Toll-like receptor 2, were major contributors in XGBoost, whereas Random Forest emphasized metabolic and mitochondrial features such as myocyte enhancer factors, lactate dehydrogenase A, ATP synthase ATP5PB, and Sirtuin 2. Across both models, large-scale connectivity differences involving salience, executive, and default mode networks (e.g., AN–SN, VN–DMN, SN–DAN) emerged as consistent drivers of classification. These findings indicate that long-term meditation experience is associated with distinct molecular and network-level adaptations that support resilience and energy regulation.

Finally, to explore links between biological features and subjective outcomes, we correlated model-identified predictors with Mystical Experience Questionnaire (MEQ) scores. Several metabolites and connectivity measures showed nominal associations ($p < 0.05$), with a subset surviving false discovery rate correction (asterisks, Fig. 8C). In the timepoint analysis, baseline 3-indolebutyric acid correlated positively with MEQ ($r = 0.72$, $q < 0.05$), whereas N6-methyladenosine correlated negatively ($r = -0.67$, q

< 0.05). At the experience level, immune markers including FGF19 and interferon- γ were negatively associated with MEQ deltas ($r = -0.59$ to -0.73 , $q < 0.05$), and reductions in network connectivity (DAN–DMN, LAI–LAG) were inversely correlated with MEQ outcomes in experienced meditators ($r = -0.64$ to -0.68 , $q < 0.05$). Together, these exploratory findings suggest that both peripheral metabolic and immune markers and central network dynamics are tied to reported mystical experience levels during meditation.”

And to the discussion at lines 522-528:

”Our machine learning analyses further reinforced these findings by demonstrating that meditation-related shifts in fMRI connectivity were embedded within broader molecular changes across metabolomic, proteomic, and transcriptomic layers. The convergence of neural and peripheral predictors, including amino acid metabolites, lipid signaling molecules, immune regulators, and network-level connectivity, underscores that meditation engages a systemic mind-body axis rather than isolated pathways.”

References:

Bremer B, et al. Mindfulness meditation increases default mode, salience, and central executive network connectivity. *Sci Rep.* 2022;12(1):13219.

Datko M, et al. Increased insula response to interoceptive attention following mindfulness training is associated with increased body trusting among patients with depression. *Psychiatry Res Neuroimaging.* 2022;327:111559.

Vishnubhotla RV, et al. Advanced Meditation Alters Resting-State Brain Network Connectivity Correlating With Improved Mindfulness. *Front Psychol.* 2021;12. <https://doi.org/10.3389/fpsyg.2021.745344>.